# Are Multiple Instance Learning Algorithms Learnable for Instances?

**Jaeseok Jang and Hyuk-Yoon Kwon**[*]
Graduate School of Data Science, Seoul National University of Science and Technology
{jangjs1027, hyukyoon.kwon}@seoultech.ac.kr

## Abstract

Multiple Instance Learning (MIL) has been increasingly adopted to mitigate the high costs and complexity associated with labeling individual instances, learning instead from bags of instances labeled at the bag level and enabling instance-level labeling. While existing research has primarily focused on the learnability of MIL at the bag level, there is an absence of theoretical exploration to check if a given MIL algorithm is learnable at the instance level. This paper proposes a theoretical framework based on probably approximately correct (PAC) learning theory to assess the instance-level learnability of deep multiple instance learning (Deep MIL) algorithms. Our analysis exposes significant gaps between current Deep MIL algorithms, highlighting the theoretical conditions that must be satisfied by MIL algorithms to ensure instance-level learnability. With these conditions, we interpret the learnability of the representative Deep MIL algorithms and validate them through empirical studies.

## 1 Introduction

The performance of supervised learning models is greatly influenced by the amount of labeled data [1]. While various models utilizing large-scale datasets have achieved excellent performance, the cost and time of labeling have emerged as issues, especially in domains requiring expert knowledge. For example, in the case of pathology images, detailed labeling requires a significant amount of expert time [2].

To address these issues, multiple instance learning (MIL) techniques have been introduced. MIL encompasses all methodologies that learn to predict the labels of instances by learning from the labels of bags composed of instances [3, 4, 5, 2, 6, 7, 8, 9, 10]. This approach allows for detecting disease areas in pathology images through whole-image labeling without requiring the labeling of regions within the image. This can significantly reduce overall labeling costs and maximize the efficiency of the method [2, 9]. As illustrated in Figure 1, if at least one instance in a bag is positive, the bag is positive, and if all instances are negative, the bag is negative in MIL. To achieve instance-level learning, it was necessary first to confirm that learning at the bag level could be performed at a high level. Therefore, traditional MIL research focused primarily on the feasibility of learning at the bag level rather than the instance level [11, 12, 13, 14], and some studies validated instance-level learning only for specific algorithms [15].

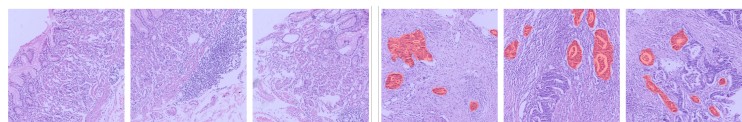

Figure 1: The data structure consisting of multi-instances (Blue: Negative, Red: Positive) [16].

Recently, with the advancement of deep learning technologies, traditional MIL has evolved into Deep MIL, enabling more effective extraction of features from individual instances and consideration of interactions between instances, leading to significant improvements in prediction performance. Despite these advancements, MIL research still predominantly focuses on learning at the bag level, with a lack of exploration into the feasibility of instance-level learning [10, 17, 18].

---

[*]Corresponding author

38th Conference on Neural Information Processing Systems (NeurIPS 2024).

In this study, we propose a new framework to theoretically validate that Deep MIL can learn at the instance level, overcoming the aforementioned issues. The contributions of this study are summarized as follows:

1. The proposed theoretical framework derives conclusions about the instance-level learnability of Deep MIL algorithms, assuming that MIL algorithms are learnable at the bag level (Assumption 1).

2. Utilizing the probably approximately correct (PAC) learning theory, we divide the hypothesis space of the dataset into two cases: 1) each instance being statistically independent and 2) the general case without any constraints on instance distributions, including statistically dependent instances. We theoretically derive the necessary and sufficient conditions for learnability in each hypothesis space (i.e., Condition 4 and Condition 7).

3. By applying the derived conditions to the existing representative types of Deep MIL, we verify their instance-level learnability.

4. Through Theorem 10 and Theorem 11, we show that additional information (e.g., positional information of each time point in time series, medical records provided alongside pathology images for better disease diagnosis) beyond the features directly extracted from the original data must act as weights in the independent hypothesis space of each instance.

This paper is organized as follows. Section 2 defines the problem. Section 3 outlines the conditions for Deep MIL to be instance learnable and provides theoretical proof. Section 4 evaluates existing Deep MIL algorithms and validates the results through experiments. Finally, Section 5 concludes the paper.

## 2  Problem Definition

**Notation 1** *(MIL domain spaces) Given the feature space for an instance $\mathcal{X}_{inst_i} \subset \mathbb{R}^d$ and the feature space for $N$ instances $\mathcal{X} := \{\mathcal{X}_{inst_1}, \mathcal{X}_{inst_2}, ..., \mathcal{X}_{inst_N}\}$, along with the label space $\mathcal{Y} := \{1, ..., k\}$, we can define the joint distribution $D_{XY}$ on $\mathcal{X} \times \mathcal{Y}$. Here, $X_{inst_i} \in \mathcal{X}_{inst_i}$, $X := (X_{inst_1}, X_{inst_2}, ..., X_{inst_N}) \in \mathcal{X}$, and $Y \in \mathcal{Y}$ are random variables. The MIL instance domain $D_{X_{inst_i}Y}$ represents the joint distribution of individual instances and their respective labels. The MIL bag domain $D_{XY}$ is the joint probability distribution composed of multiple instance domains $D_{X_{inst_i}Y}$. Here, the term "domain" refers to these joint distributions.*

Based on Notation 1, the MIL Problem can be defined as in Definition 1.

**Definition 1** *(MIL problem) Given a training dataset $S := \{((x_1^1, ..., x_n^1), y^1), ..., ((x_1^m, ..., x_n^m), y^m)\}$ drawn IID from the joint distribution $D_{XY}$, the goal of MIL is to learn a classifier $f_{bag}$ for the data and $(f_{inst_1}, ..., f_{inst_n})$ such that for any arbitrary bag random variable $x := (x_1, ..., x_n)$ drawn from the marginal distribution $D_X$: 1) It should be able to classify the class corresponding to the bag $x$. 2) It should be able to classify the class corresponding to the $i^{th}$ instance $x_i$ of the bag. Here, a random sample $(x, y) := ((x_1, ..., x_n), y)$ of the training data represents a bag consisting of $n$ instances drawn from $D_{XY}$, with a total of $m$ bags. The training data for the $i^{th}$ instance is given by $S_{inst_i} := \{(x_i^1, y^1), ...(x_i^m, y^m)\}$.* □

The MIL problem defined in Definition 1 is performed within the following MIL hypothesis spaces.

**Notation 2** *(MIL hypothesis spaces) 1) The bag hypothesis space $\mathcal{H}_{bag} \subset \{h_{bag} : X \rightarrow Y\}$ is a set of hypothesis functions $h_{bag}$ that classify bags with the correct labels. Here, $h_{bag} = f(X)$, where $f \in \mathcal{F}_{bag}$. 2) The $i^{th}$ instance hypothesis space $\mathcal{H}_{inst_i} \subset \{h_{inst_i} : X_{inst_i} \rightarrow Y\}$ is a set of hypothesis functions $h_{inst_i}$ that classify the $i^{th}$ instance with the correct labels using the feature space of the $i^{th}$ instance. Here, $h_{inst_i} = f(X_{inst_i})$, where $f \in \mathcal{F}_{inst_i}$. Here, $\mathcal{F}_{bag}$ and $\mathcal{F}_{inst_i}$ are sets of functions that take a bag or the $i^{th}$ instance as input and generate the corresponding labels.*

According to Notations 1, 2, and Definition 1, the risk for a bag in MIL is defined as in Definitions 2 and 3.

**Definition 2** *(Bag Risk) If the loss for a bag in the MIL algorithm is given by $\ell_{bag}(h_{bag}(x), y)$, the risk for a bag is defined as follows:*

$$R_{bag} = \mathbb{E}_{(x,y) \sim D_{XY}} \ell_{bag}(h_{bag}(x), y) \tag{1}$$

**Definition 3** *(Instance Risk) If the loss for the $i^{th}$ instance in the MIL algorithm is given by $\ell_{inst_i}(h_{inst_i}(x_i), y)$, the risk for the $i^{th}$ instance is defined as follows:*

$$R_{inst_i} = \mathbb{E}_{(x_i,y) \sim D_{X_{inst_i}Y}} \ell_{inst_i}(h_{inst_i}(x_j), y) \tag{2}$$

Based on the definitions of bag risk and instance risk in Definition 2 and 3, the learnability for bags and instances can be defined as in Definition 4 and Definition 5.

**Definition 4** *(PAC Learnability of Bag) Given the domain space $\mathcal{D}_{XY}$ and the bag hypothesis space $\mathcal{H}_{bag} \subset \{h_{bag} : \mathcal{X} \to \mathcal{Y}\}$, the MIL algorithm $A$ is said to be learnable on $\mathcal{H}_{bag}$ with respect to $\mathcal{D}_{XY}$ if, for all domains $D_{XY} \in \mathcal{D}_{XY}$, the following condition is satisfied:*

$$\mathbb{P}_{S \sim D_{XY}^m}[|R_{bag}(A(S)) - \inf_{h \in \mathcal{H}_{bag}} R_{bag}(h)| \le \epsilon] \ge 1 - \delta \tag{3}$$

*Here, $\epsilon$ represents the acceptable error between the learning algorithm and the actual optimal hypothesis. In contrast, $\delta$ represents the confidence level that the learning algorithm will return accurate results within a certain error range. Both $\epsilon$ and $\delta$ have a range of $0 < \epsilon, \delta < 1$.*

**Definition 5** *(PAC Learnability of Instance) When given the $i^{th}$ instance domain space $\mathcal{D}_{X_{inst_i}Y}$ and instance hypothesis space $\mathcal{H}_{inst_i} \subset \{h_{inst_i} : X \to Y\}$, the algorithm $A$ is said to be learnable over $\mathcal{H}_{inst_i}$ from $\mathcal{D}_{X_{inst_i}Y}$ if it satisfies the following for all domains $D_{X_{inst_i}Y} \in \mathcal{D}_{X_{inst_i}Y}$:*

$$\mathbb{P}_{S_{inst_i} \sim D_{X_{inst_i}Y}^m}[|R_{inst_i}(A(S_{inst_i})) - \inf_{h \in \mathcal{H}_{inst_i}} R_{inst_i}(h)| \le \epsilon] \ge 1 - \delta \tag{4}$$

Based on the relationship between Definitions 4 and 5, according to Theorem 1, if a MIL algorithm is not learnable with respect to bags, it is not learnable with respect to instances.

**Theorem 1** *If MIL algorithm $A$ satisfies Condition 1, then this algorithm is not PAC learnable for any instance domain space $\mathcal{D}_{X_{inst_i}Y}$ and instance hypothesis space $\mathcal{H}_{inst_i} \subset \{h_{inst_i} : X \to Y\}$.*

$$\mathbb{P}\left[\bigcup_{i=1}^{n} |R_{inst_i}(A(S_{inst_i})) - \inf_{h \in \mathcal{H}_{inst_i}} R_{inst_i}(h)| > \epsilon\right] > \delta \tag{5}$$

***Condition 1*** *The MIL algorithm $A$ is not PAC learnable for the given domain space $\mathcal{D}_{XY}$ and bag hypothesis space $\mathcal{H}_{bag} \subset \{h_{bag} : X \to Y\}$:*

$$\mathbb{P}\left[|R_{bag}(A(S)) - \inf_{h \in \mathcal{H}_{bag}} R_{bag}(h)| > \epsilon\right] > \delta \tag{6}$$

***Proof***: *The proof of Theorem 1 is conducted in Appendix C.1.*

According to Theorem 1, MIL algorithms that are not learnable for bags do not guarantee learnability for instances. Therefore, in this study, we discuss the learnability of instances under Assumption 1.

**Assumption 1** *The MIL algorithm is PAC learnable for bags.*

Based on the definitions, theorem, and assumption above, we can formulate the definition of when the proposed Deep MIL is learnable for instances as follows:

**Definition 6** *If the MIL algorithm satisfies Condition 2, it is learnable for instances.*

**Condition 2** *The Deep MIL algorithm A must exhibit equivalent PAC learnability for bags and instances:*

$$\mathbb{P}\left[|R_{bag}(A(S)) - \inf_{h \in \mathcal{H}_{bag}} R_{bag}(h)| \leq \epsilon \wedge \bigcap_{i=1}^{n} |R_{insti}(A(S_{inst_i})) - \inf_{h \in \mathcal{H}_{inst_i}} R_{inst_i}(h)| \leq \epsilon\right] \geq 1-\delta \tag{7}$$

Definition 6 under Assumption 1 ensures that if Condition 2 is satisfied, the algorithm is guaranteed to be learnable for instances. On the other hand, MIL algorithms that do not satisfy Condition 2 cannot guarantee learnability for instances, even if they successfully learn for bags. Therefore, Condition 2 becomes a necessary and sufficient condition for MIL algorithms to be learnable for instances.

## 3 Proposed Theoretical Framework

In this study, we propose a theoretical framework to verify whether a given MIL algorithm satisfies Condition 2 according to Definition 6. For some Deep MIL algorithms [5, 6], instances are assumed to belong to independent bag domain spaces. Therefore, we address the problem by distinguishing between the independent bag domain space $\mathcal{D}_{XY}^{Ind}$ and the general bag domain space $\mathcal{D}_{XY}^{Gen}$. 1) $\mathcal{D}_{XY}^{Ind}$ refers to bag domain spaces where all instances within a bag are statistically independent. 2) $\mathcal{D}_{XY}^{Gen}$ refers to a bag domain space that includes both $\mathcal{D}_{XY}^{Dep}$, where interactions or dependencies exist among instances within a bag, and $\mathcal{D}_{XY}^{Ind}$. That is, $\mathcal{D}_{XY}^{Ind} \cup \mathcal{D}_{XY}^{Dep} = \mathcal{D}_{XY}^{Gen}$.

### 3.1 Overview

Figure 2 shows the final summary of the definitions, relationships, and results of the theorems that comprise the theoretical framework proposed in this study.

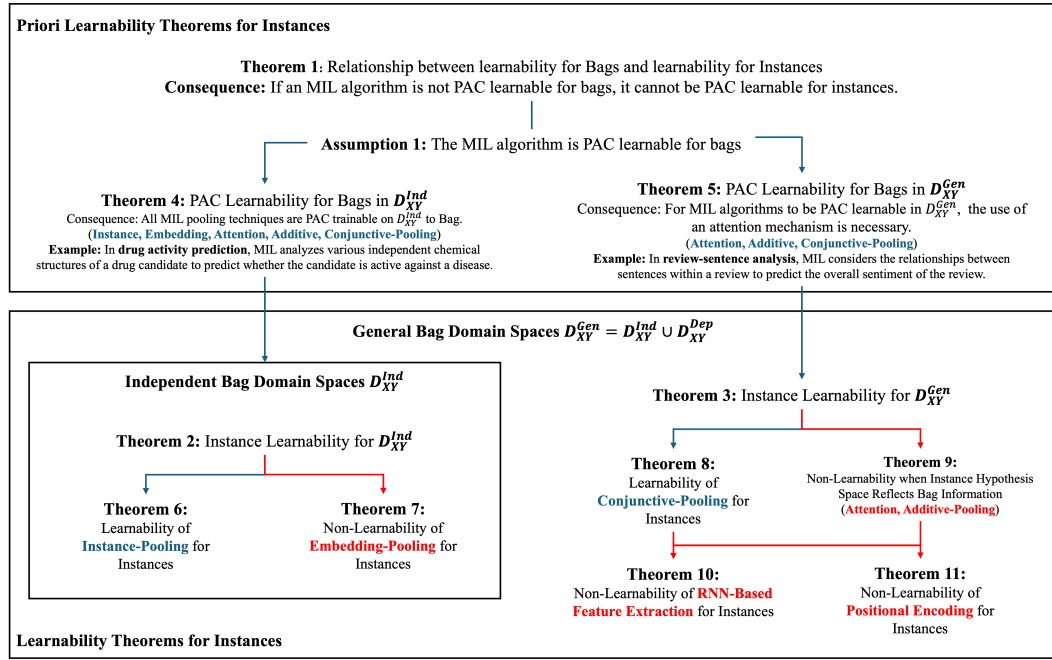

Figure 2: Relationships between theorems: Blue arrows indicate that the pooling methods are learnable when our proposed conditions are satisfied; Red arrows indicate that they are not learnable when the conditions are not satisfied.

### 3.2 PAC Learnability for Independent Bag Domain Spaces

The definition of $\mathcal{D}_{XY}^{Ind}$ is provided in Definition 7.

**Definition 7** *The independent Bag Domain Space $\mathcal{D}_{XY}^{Ind}$ is defined as a domain Space that encompasses all instance spaces while satisfying Condition 3:*

**Condition 3** *For each Instance Space $D_{X_{inst_i} Y}$ within $\mathcal{D}_{XY}^{Ind}$, there must exist a corresponding bag domain space $D_{XY} \in \mathcal{D}_{XY}^{Ind}$, which should be determined as the union of each instance domain space, as follows:*

$$D_{XY}^{Ind} := \bigcup_{i=1}^{N} D_{X_{inst_i} Y} \in \mathcal{D}_{XY}^{Ind} \tag{8}$$

Since $D_{XY}^{Ind}$ satisfies Condition 3, implying independence among instances, the hypothesis space for each instance is unaffected by other instances. In this case, for MIL algorithms to be learnable for instances, they must satisfy Condition 4 according to Theorem 2.

**Theorem 2** *If a MIL algorithm satisfies Condition 4 in $\mathcal{D}_{XY}^{Ind}$, it is learnable for instances.*

**Condition 4** *The risk of the optimal hypothesis for $D_{XY}^{Ind}$ must ensure that it equals the sum of the risks of the optimal hypotheses for individual instance spaces within $D_{XY}^{Ind}$:*

$$\inf_{h \in \mathcal{H}} R_{\mathcal{D}_{XY}^{Ind}} = \sum_{i=1}^{N} \inf R_{inst_i} \tag{9}$$

*Consequence*: *If Condition 4 is satisfied, then Condition 2 is also satisfied. Hence, Condition 4 becomes a necessary and sufficient condition for learnability for instances in $D_{XY}^{Ind}$.*

**Proof**: *The proof of Theorem 2 is provided in Appendix C.2.*

### 3.3 PAC Learnability for General Bag Domain Spaces

The definition of $\mathcal{D}_{XY}^{Gen}$ is provided in Definition 8.

**Definition 8** *The general bag domain space $\mathcal{D}_{XY}^{Gen}$ is defined as a domain space that encompasses all instance spaces while satisfying Condition 5 and 6:*

**Condition 5** *For every instance space $\mathcal{D}_{X_{inst_i} Y}$ within $\mathcal{D}_{XY}^{Gen}$, there exists a corresponding bag domain space $D_{XY} \in \mathcal{D}_{XY}^{Gen}$, determined as the sum of each instance domain space.*

**Condition 6** *$\mathcal{D}_{XY}^{Gen}$ is formed using weights $\alpha_i \in (0, 1)$ to reflect the importance of relationships among instances. Each instance domain space $D_{X_{inst_i} Y}$ should be defined along with its weight $\alpha_i$ as follows:*

$$D_{XY}^{Gen} = \sum_{i=1}^{N} \alpha_i D_{X_{inst_i} Y} \in \mathcal{D}_{XY}^{Gen} \quad such \ that \quad \sum_{i=1}^{N} \alpha_i = 1, \quad 0 \le \alpha_i \le 1 \tag{10}$$

Since $\mathcal{D}_{XY}^{Gen}$ satisfies Condition 5 and Condition 6, implying the existence of relationships among instances, the hypothesis space for each instance is influenced by other instances. In this case, for MIL algorithms to be learnable for instances, they must satisfy Condition 7 according to Theorem 3.

**Theorem 3** *If a MIL algorithm satisfies Condition 7 in $\mathcal{D}_{XY}^{Gen}$, it is learnable for instances.*

**Condition 7** *The risk of the optimal hypothesis for $D_{XY}^{Gen}$ must ensure that it equals the weighted sum of the risks of the optimal hypotheses for individual instance spaces within $D_{XY}^{Gen}$:*

$$\inf_{h \in \mathcal{H}} R_{\mathcal{D}_{XY}^{Gen}} = \sum_{i=1}^{N} \alpha_i \inf R_{inst_i} \quad such \ that \quad \sum_{i=1}^{N} \alpha_i = 1, \quad 0 \le \alpha_i \le 1 \tag{11}$$

*Consequence*: *If a MIL algorithm satisfies Condition 7, it also satisfies Condition 2. Hence, Condition 7 becomes a necessary and sufficient condition for MIL algorithms to be learnable for instances in $\mathcal{D}_{XY}^{Gen}$.*

**Proof**: *The proof of Theorem 3 is provided in Appendix C.3.*

# 4 Theoretical Verification of Existing Deep MILs

## 4.1 Classifications of Existing Deep MIL Methodologies

The Deep MIL algorithms proposed so far are primarily categorized based on 1) whether they perform Aggregation at the Embedding-level [5, 6, 2, 19, 20] or at the Instance-level [5, 21, 22, 23, 9, 8, 10, 24]. Additionally, they can be further classified into 5 types of pooling techniques according to 2) whether they do not use an attention mechanism [5, 21, 22, 6], perform Aggregation by multiplying attention weights at the embedding-level [2, 19, 20], or perform aggregation by multiplying attention weights at the instance-level [23, 9, 10, 24], as shown in Table 1.

Table 1: Classification of existing Deep MIL methodologies.

|  | Instance -pooling [5, 21] | Embedding -pooling [5, 6] | Attention -pooling [2, 19] | Additive -pooling [9] | Conjunctive -pooling [23, 10, 24] |
|---|---|---|---|---|---|
| Aggregation-level | Instance | Embedding | Embedding | Instance | Instance |
| Attention-target | None | None | Embedding | Embedding | Instance |

A detailed explanation of the pooling techniques is provided in Appendix A.2.

## 4.2 Theoretical Verification

### 4.2.1 Relationship between Attention Mechanism and Learnability for Bag

The application of an attention mechanism to MIL algorithms depends on the range of the domain space that the MIL algorithm can learn from. This implies that the feasibility of PAC learning for bags may vary depending on whether attention is applied or not.

**Theorem 4** *In $\mathcal{D}_{XY}^{Ind}$, the MIL algorithm is PAC learnable for bags.*

**Proof**: *The proof of Theorem 4 is conducted in Appendix C.4.*

**Theorem 5** *In $\mathcal{D}_{XY}^{Gen}$, MIL algorithms that aggregate independent hypothesis spaces for each instance must utilize attention satisfying Condition 8 to be PAC learnable for bags.*

**Condition 8** *The hypothesis space $\mathcal{H}_{bag_{att}}$ of Attention MIL should be equal to the sum of independent hypothesis spaces $h_{inst_i}$ multiplied by attention weights $Att_i$:*

$$\mathcal{H}_{bag_{att}} = \left\{ h_{bag_{att}} \mid h_{bag_{att}} = \sum_{i=1}^{n} Att_i \cdot h_{inst_i}, \; where \; 0 < Att_i < 1, \; \sum_{i=1}^{n} Att_i = 1 \right\} \quad (12)$$

**Proof**: *The proof of Theorem 5 is conducted in Appendix C.5.*

**Validation**: *Experimental validation is performed in Section 4.4.1.*

**Consequence:** *Theorems 4 and 5 are not direct theorems about the learnability of MIL for instances. However, the learnability for bags is a prerequisite for MIL algorithms to be able to learn from instances. According to Theorem 4, in PAC learnable $\mathcal{D}_{XY}^{Ind}$, all MIL algorithms satisfy Assumption 1 regardless of the presence of attention mechanisms. On the other hand, in $\mathcal{D}_{XY}^{Gen}$, according to Theorem 5, the application of attention mechanisms satisfying Condition 8 becomes a necessary condition for MIL algorithms to be learnable for instances.*

### 4.2.2 Verification Learnability for Instances by MIL Pooling Method

The type of pooling technique used in Deep MIL algorithms becomes a determining factor in whether they are learnable for instances when they are learnable for bags. In this case, to verify whether a Deep MIL algorithm is learnable for instances, a definition of Lemma 1 should be established in advance, extending Condition 4 to ensure that no additional hypothesis space is included for instances.

**Lemma 1** *Condition 9 serves as a necessary condition for the learnability of instances, when the hypothesis space for the $i^{th}$ instance of a MIL algorithm is $\mathcal{H}_{inst_i} \cup \mathcal{H}_{add_i}$. Here, $\mathcal{H}_{inst_i}$ represents the hypothesis space for the $i^{th}$ instance, and $\mathcal{H}_{add_i}$ denotes the hypothesis space for the $i^{th}$ instance generated through elements outside the $i^{th}$ instance.*

**Condition 9** $\mathcal{H}_{add_i}$ *must be a subset of* $\mathcal{H}_{inst_i}$:

$$\mathcal{H}_{inst_i} \supset \mathcal{H}_{add_i} := \{h_{add_i} : \mathcal{X}_{add_i} \to \mathcal{Y}\} \tag{13}$$

**Proof**: *The proof of Lemma 1 is conducted in Appendix C.6.*

In the case of instance-pooling, as attention mechanisms are not utilized, Condition 8 is not satisfied. Therefore, in $\mathcal{D}_{XY}^{Gen}$, the algorithm is not learnable for bags, and consequently, not learnable for instances either. However, according to Theorem 6, the algorithm is learnable for instances in $\mathcal{D}_{XY}^{Ind}$.

**Theorem 6** *In $\mathcal{D}_{XY}^{Ind}$, MIL algorithms that perform instance-pooling are PAC learnable for instances.*

**Proof**: *The proof of Theorem 6 is conducted in Appendix C.7.*

Unlike Instance-Pooling, where the hypothesis for bags is combined from instance-level hypotheses, Embedding-Pooling does not combine bag hypotheses from instance-level hypotheses. As a result, it does not satisfy Condition 9, leading to a scenario similar to Theorem 7.

**Theorem 7** *MIL algorithms that perform Embedding-Pooling are not learnable for instances.*

**Proof**: *The proof of Theorem 7 is conducted in Appendix C.8.*

**Consequence:** *In $\mathcal{D}_{XY}^{Ind}$, Deep MIL algorithms exhibit reproducibility when they avoid overfitting to easy information at the bag-level and effectively learn positive instances at the instance-level. Reproducibility in this context refers to the learnability of positive instances. They demonstrated experimentally that Deep MIL algorithms using Instance-Pooling, such as mi-Net [5] and Causal MIL [21, 22], exhibit reproducibility, while those using Embedding Pooling, such as Mi-Net [5], do not. However, Raff et al. [18] failed to provide a theoretical explanation for these results. In contrast, this study theoretically demonstrated that Instance-Pooling is learnable for instances while Embedding-Pooling is not, through Theorems 6 and 7. This provides theoretical support for the experimental findings of Raff et al. [18].*

To compute the attention applied to each instance's features in Attention Pooling and Additive Pooling, the bag's features $X$ are used as input. Multiplying attention weights to the features at the feature level results in additional hypothesis space $h_{add_i}$ formed by the attention operations on each instance. As a consequence, since Condition 9 is not satisfied, according to Theorem 8, the algorithm becomes not PAC Learnable for instances.

**Theorem 8** *If the MIL algorithm does not adhere to Condition 10, it is not learnable for instances.*

**Condition 10** *The risk $R_{inst_i}$ for the $i^{th}$ instance should be as follows:*

$$R_{inst_i} = \mathbb{E}_{(x_{inst_i}, y) \sim D_{X_{inst_i} Y}} \ell_{inst_i}(h, y) \quad, where \quad h \in \mathcal{H}_{inst_i} \cup \mathcal{H}_{bag-level_i} \tag{14}$$

*Here, $\mathcal{H}_{bag-level_i}$ denotes the hypothesis space for the $i^{th}$ instance generated through bag-level features, and $\mathcal{H}_{bag-level_i} := \{h_{bag-level} : \mathcal{X} \to \mathcal{Y}\}$.*

**Consequence**: *Attention-Pooling and Additive-Pooling multiply attention weights to each instance's feature-level. As a result, the hypothesis space for the $i^{th}$ instance includes, in addition to Instance-Pooling, $h_{bag-level} = \{h \mid h = f(X_{bag}), f \in \mathcal{F}_{bag}\}$. In other words, it incorporates the hypothesis space for bag-level features into the prediction for the $i^{th}$ instance. This, according to Theorem 8, renders it not learnable for instances.*

**Proof**: *The proof of Theorem 8 is conducted in Appendix C.9.*

**Validation**: *Experimental verification of Theorem 8 is demonstrated in Section 4.4.2.*

On the other hand, Conjunctive-Pooling does not multiply attention at the feature level of instances, but rather at the prediction level of instances. Therefore, predictions are made based on individual features of instances, resulting in $\mathcal{H}_{bag-level_i} = \emptyset$, satisfying Condition 10. Additionally, Theorem 9 demonstrates that Conjunctive-Pooling operates in a manner that is learnable for instances.

**Theorem 9** *When MIL algorithms use Conjunctive-Pooling for aggregation in $\mathcal{D}_{XY}^{Gen}$, they are learnable for instances.*

***Consequence:*** *According to Theorem 9, Conjunctive-Pooling becomes the unique pooling technique learnable for instances in $\mathcal{D}_{XY}^{Gen}$. Specifically, since $\mathcal{D}_{XY}^{Gen}$ includes $\mathcal{D}_{XY}^{Ind}$ by definition, Conjunctive-Pooling becomes a methodology that satisfies all cases.*

***Proof***: *The proof of Theorem 9 is conducted in Appendix C.10.*

***Validation***: *Experimental validation for Theorem 9 is presented in Section 4.4.2.*

Javed et al.[9] demonstrated that the contribution of instances in MIL algorithms performing Additive-Pooling is proportional to the shapley value[25]. However, their proof contained an error where the feature multiplied by attention was mistakenly assumed to be the feature of the $i^{th}$ instance. Our theoretical framework identifies that when attention is applied to features, it fails to satisfy Condition 10, leading to the algorithm being not learnable.

In this study, we confirmed that the error causing failure to satisfy Condition 10 also appears in studies proposing MIL based on Conjunctive-Pooling [23, 10]. Details on this are explained in Section 4.3.1.

### 4.3 Additional Considerations

### 4.3.1 Rethinking Position Dependencies of Instances on Deep MILs

Text data and time series data exhibit temporal dependencies, while image data often has spatial dependencies. Therefore, research in relevant fields has utilized neural networks capable of capturing dependencies, such as RNNs and CNNs [26, 27, 28, 29, 30, 31], or additional positional encoding on extracted features to enhance performance [32, 33, 34, 35, 36, 37]. Following this trend, Deep MIL studies have also employed RNN-based neural networks or positional encoding during the feature extraction process to capture temporal dependencies of instances for performance enhancement [23, 19, 10]. However, as per Theorem 10 and Theorem 11, these approaches render the models unable to learn from instances.

**Theorem 10** *If the MIL algorithm extracts features of instances through RNN-based neural networks for aggregation, it is unable to learn from instances.*

***Proof***: *The proof for Theorem 10 is conducted in Appendix C.11.*

***Validation***: *The experimental validation for Theorem 11 is presented in Section 4.4.3.*

**Theorem 11** *If the hypothesis space $\mathcal{H}_{Pos-Encode_i}$ generated through positional encoding values for the $i$-th position of the MIL algorithm is not a subset of $\mathcal{H}_{inst_i}$, then the algorithm is not PAC learnable for instances.*

***Proof***: *If $\mathcal{H}_{Pos-Encode_i} \not\subset \mathcal{H}_{inst_i}$, the algorithm fails to satisfy Condition 9, rendering it not learnable for instances.*

***Validation***: *Experimental validation for Theorem 10 is shown in Section 4.4.3.*

***Consequence:*** *According to Theorem 11, using values outside of an instance's features in the prediction process makes it unlearnable. Therefore, positional encoding should not be used in the process of predicting instances.*

According to the investigation conducted in this study, it was observed that existing Deep MIL research using Conjunctive-Pooling, such as MILNET [23] and MILLET [10], all utilized RNN-based neural networks for feature extraction or performed positional encoding on instance features. Therefore, they failed to satisfy Condition 9, making them unable to learn about instances. Furthermore, these theoretical findings provide a basis for the ablation study conducted by Early et al.[10] on time series data classification problems, where Conjunctive-Pooling and positional encoding were applied to predict the class at each time step. In essence, Early et al.[10] demonstrated that while positional encoding contributed to improving prediction performance at the bag level, it acted as a detrimental factor in predicting instances. When utilizing external information such as positional dependencies through supplementary weighting factors like attention operations alone, Condition 9 may be satisfied, thereby enabling learnability concerning instances.

### 4.3.2 Learnability for Instances in Each Dimension for Multidimensional Deep MILs

In the real world, data is often composed of bags consisting of multi-dimensional instances rather than simple instances of uniform dimensions. For instance, in video data, each frame(images)

is composed of patches in multi-dimensional structures for each frame dimension. Similarly, in multivariate time series data, each timestamp is composed of multivariate data points. In the case of multi-dimensional instances, Multi-dimensional Deep MIL (MD-MIL) methodologies have emerged to perform predictions on lower-level instances based solely on labels at the top level of the bag. These methodologies apply aggregation recursively, performing aggregation on data in the first dimension and sequentially on subsequent dimensions. Existing MD-MIL approaches have employed Embedding-Pooling or Attention-Pooling for each dimension's instances. However, these pooling methods have been shown by Theorems 7 and 8 to be incapable of learning about instances. Therefore, existing MD-MIL methodologies are not suitable for learning from multi-dimensional instances.

Therefore, to design methodologies capable of learning from multi-dimensional data, it is essential to use aggregation techniques that are capable of learning about instances, based on Theorem 6 or 9. Additionally, to ensure learnability from the top-level bag and bags in each dimension, the attention operation's results should be set to be within the instance's hypothesis space based on whether the relationships among instances in each dimension are independent or dependent. Through the experiments in Appendix E.2, we confirmed that Conjunctive-Pooling can capture the relationships between instances across different bags in an MD-MIL architecture, leading to performance improvements. These findings confirm that our proposed theoretical framework serves as a valuable guide in designing learnable models.

## 4.4 Experimental Validation

In this section, we conduct the following experimental validations to demonstrate whether existing Deep MIL approaches are learnable for instances based on the theorems: 1) **(Theorem 5)**: Demonstrating the learnability of the attention mechanism for bags in $\mathcal{D}_{XY}^{Gen}$. 2) **(Theorem 8, 9)**: Showing that multiplying attention at the feature level is not learnable for instances. 3) **(Theorem 10, 11)**: Demonstrating that inputting position-related values into instance positions is not learnable for instances. As this study assumes an environment where bags are PAC Learnable, we preprocess the MNIST dataset to match the difficulty level of each experiment. For the validation of Theorem 10 and 11, we use the WebTraffic dataset from Early et al. [10], which is a synthetic time-series classification dataset. Detailed experimental settings can be found in Appendix D.

### 4.4.1 Experimental Validation of Theorem 5

To validate Theorem 5, we conducted experiments on MIL algorithms [5, 21, 38, 2, 39, 40, 19, 9, 10] representing each pooling technique. Table 2 compares prediction performance on the synthetic datasets, detailed in Appendix D.1. This reveals that Instance-Pooling based MIL algorithms [5, 21, 38], which do not apply weights to the hypothesis space, degraded learning performance for bags in $\mathcal{D}_{XY}^{Gen}$. In contrast, the other algorithms [2, 39, 40, 19, 9, 10] that apply weights to the hypothesis space through the attention mechanism demonstrate superior performance, which is even comparable to a none-pooling-based method that, by using fully connected layers, preserves all instance-level information without any loss during prediction. This experimentally validates Theorem 5.

Table 2: Prediction performance of Deep MIL on Bags in $D_{XY}^{Gen}$.

| Pooling Methods | Deep MIL Algorithms | Macro-F1 | Micro-F1 | Weighted-F1 |
|---|---|---|---|---|
| Instance-Pooling | mi-Net [5] | 0.3286 | 0.5548 | 0.4550 |
| | Causal MIL [21] | 0.2341 | 0.3577 | 0.2645 |
| | MIREL [38] | 0.3623 | 0.5318 | 0.4372 |
| Attention-Pooling | Attention MIL [2] | 0.7652 | 0.7683 | 0.7583 |
| | Loss-Attention [39] | 0.7935 | 0.7832 | 0.7753 |
| | SA-AbMILP [40] | 0.7540 | 0.7619 | 0.7562 |
| | TransMIL [19] | 0.7834 | 0.7711 | 0.7738 |
| Additive-Pooling | Additive MIL [9] | 0.5314 | 0.6341 | 0.5732 |
| Conjunctive-Pooling | Conjunctive MIL [10] | 0.7544 | 0.7701 | 0.7683 |
| None-Pooling | Fully Connected | 0.7704 | 0.7724 | 0.7714 |

#### 4.4.2 Experimental Validation of Theorems 8 and 9

To assess whether the algorithm [2, 39, 40, 19, 9, 10] is learnable for instances when weights from an attention mechanism that satisfies Condition 8 are multiplied at the feature level, we compared the predictive performance on instances by adjusting the variance of attention weights.

Table 3 shows the results on the synthetic dataset, which details in Appendix D.2. Attention-Pooling based [2, 39, 40, 19] and Additive-Pooling based [9] MIL algorithms, where attention weights are multiplied at the feature level and aggregation is performed, showed significantly lower predictive performance for instances compared to bags. In particular, SA-AbMILP [40] and TransMIL [19], which perform iterative attention operations, show a significant performance gap between bag-level predictions and instance-level predictions. This demonstrates that the Attention-Pooling process does not guarantee the learnability of MIL. In contrast, Conjunctive-Pooling based MIL algorithms [10] exhibited a much smaller difference in predictive performance between bags and instances than other algorithms. This validates, in accordance with Theorem 9, that Conjunctive-Pooling is learnable for instances. Additionally, experiments adjusting the variance of attention to determine its impact on the discrepancy between bag and instance performance are detailed in Appendix E.1.

Table 3: Prediction performance comparison of MIL algorithms on bags and instances.

|  | Performance for bags ($P_{Bag}$) | | Performance for Instances ($P_{Inst}$) | | $P_{Inst} - P_{Bag}$ | |
| --- | --- | --- | --- | --- | --- | --- |
| Deep MIL Algorithms | Macro-F1 | AUROC | Macro-F1 | AUROC | Macro-F1 | AUROC |
| Attention MIL [2] | 0.8434 | 0.9516 | 0.3215 | 0.7317 | -0.5219 | -0.2199 |
| Loss-Attention [39] | 0.8228 | 0.9574 | 0.4797 | 0.7951 | -0.3431 | -0.1623 |
| SA-AbMILP [40] | 0.7692 | 0.9552 | 0.3340 | 0.5464 | -0.4352 | -0.4088 |
| TransMIL [19] | 0.8515 | 0.9622 | 0.2192 | 0.5369 | -0.6323 | -0.4253 |
| Additive MIL [9] | 0.4776 | 0.9181 | 0.2320 | 0.8092 | -0.2456 | -0.1089 |
| **Conjunctive MIL** [10] | 0.7916 | 0.9463 | 0.6430 | 0.9516 | **-0.1486** | **+0.0053** |

#### 4.4.3 Experimental Validation of Theorems 10 and 11

Table 4 compares the performance of Conjunctive MIL under various conditions, reflecting information on positional dependency: 1) Applying attention and prediction on features extracted via Positional Encoding or a GRU-layer (All), 2) Using positional information only for attention operations (Att), 3) Using positional information only for prediction (Predict), and 4) The baseline model without any positional adaptations (Default).

Comparative results on the WebTraffic dataset's instance prediction performance reveal that configurations All and Predict, which provide additional positional information to features as in Theorem 10 and 11, showed poorer prediction performance than Default. Particularly, RNN, which reflects more additional information than positional encoding, was found to significantly degrade performance. However, the Att configuration, which utilized positional information solely for attention operations, achieved better performance than Default. This demonstrates that incorporating helpful information such as positional data, into MIL should be selectively applied to attention computations only.

Table 4: Test positional dependencies for WebTraffic datasets [10].

|  | Default | PE (All) | PE (Att) | PE (Predict) | RNN (All) | RNN (Att) | RNN (Predict) |
| --- | --- | --- | --- | --- | --- | --- | --- |
| AOPCR | 13.041 | 12.372 | 14.555 | 12.256 | 9.011 | **17.502** | 12.21 |
| NDCG@n | 0.676 | 0.665 | **0.727** | 0.642 | 0.620 | 0.714 | 0.523 |

## 5 Conclusions

In this study, we proposed a novel framework to theoretically validate that deep MIL can learn at the instance level, overcoming the aforementioned issues. The proposed theoretical framework derived conclusions about the instance-level learnability of deep MIL algorithms, assuming that MIL algorithms are learnable at the bag level. Utilizing the PAC learning theory, we theoretically derived the necessary and sufficient conditions for learnability in each hypothesis space. This provides theoretical guidance for building learnable MIL models in various domains. The practical application of our proposed framework to real-world MIL scenarios is presented in Appendix F. Limitations and future work are explained in Appendix G.

## Acknowledgement

This work was supported by the National Research Foundation of Korea(NRF) grant funded by the Korea government(MSIT) (No. 2022R1F1A1067008), and by the Basic Science Research Program through the National Research Foundation of Korea(NRF) funded by the Ministry of Education (No. 2019R1A6A1A03032119).

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

# A   Related Works

## A.1   Multiple Instance Learning

Multiple Instance Learning (MIL), a type of weakly supervised learning, is a methodology where labels are assigned to a bag of instances rather than individual instances, allowing for predictions about individual instances based on the bag's label. MIL was first introduced in the field of chemistry to identify active molecules and has since been applied to various domains, including medical image analysis, text data processing, time-series data analysis, and video anomaly detection [3, 4, 41, 2, 9, 42, 23, 43, 44, 10, 45, 7, 8].

In medical image analysis, accurate labeling of cancer cell locations requires significant time and effort. However, it is possible to use MIL to predict cancer cell locations with labels for the entire image. In text data, documents or reviews are treated as a single bag, and MIL allows for more detailed analysis by predicting the attributes of each instance within a document or review. MIL is also effectively used in time-series data and video anomaly detection for cause analysis and anomaly detection.

## A.2   Type of Pooling in Multiple Instance Learning

Deep MIL methodologies can be broadly classified based on the pooling mechanism into the following categories: 1) Instance-Pooling, 2) Embedding-Pooling, 3) Attention-Pooling, 4) Additive-Pooling, and 5) Conjunctive-Pooling.

The functional architecture of MIL incorporates several key components designed for processing instances within a bag. The feature extraction function for the $i^{th}$ instance is denoted by $f_i(X_i)$. Furthermore, we utilize classifier functions to make predictions based on these features: $p_i(X_i)$ targets the features of the $i^{th}$ instance, while $p(X)$ handles features aggregated at the bag level. The attention weight assigned to the $i^{th}$ instance is expressed as $A_i$. Additionally, the aggregation function, pivotal for our pooling techniques, is represented by $g(X)$. The various pooling methods utilized in our framework are described as follows:

**(Instance-Pooling)**

Instance-pooling performs predictions for each instance individually and then uses the max or mean operator to aggregate the results:

$$g_{max-pooling}(X) = \max_{i=1}^{N} (p_i \circ f_i)(X_i)$$

$$g_{mean-pooling}(X) = \frac{1}{N} \sum_{i=1}^{N} (p_i \circ f_i)(X_i)$$

**(Embedding-Pooling)**

Embedding-pooling obtains features for each instance and then uses the max or mean operator to aggregate these features to obtain a feature representation for the bag, on which predictions for the bag are made:

$$g_{max-pooling}(X) = p(\max_{i=1}^{N} f_i(X_i))$$

$$g_{mean-pooling}(X) = p(\frac{1}{N} \sum_{i=1}^{N} f_i(X_i))$$

**(Attention-Pooling)**

Attention-pooling multiplies the features of each instance by attention weights and then performs embedding-pooling using the summation operator on the weighted features:

$$g(X) = p(\sum_{i=1}^{N} A_i f_i(X_i))$$

**(Additive-Pooling)**

Additive-pooling multiplies the features of each instance by attention weights, obtains individual predictions for each instance, and then performs a summation operation to make predictions for the bag:

$$g(X) = \sum_{i=1}^{N} (p_i \circ A_i f_i)(X_i)$$

**(Conjunctive-Pooling)**

Conjunctive-pooling multiplies the individual predictions of instances by attention weights and performs aggregation through a weighted sum:

$$g(X) = \sum_{i=1}^{N} (A_i p_i \circ f_i)(X_i)$$

### A.2.1 Attention Mechanism

In early deep MIL, the role of the deep network was to better extract the features of instances and to accurately predict instances through pooling operations in an end-to-end manner [5, 21, 22, 6, 46, 47, 48].

Early deep MIL models had the limitation that the bag-level prediction does not accelerate the instance-level prediction because predictions for instances were performed separately from predictions for the bag. To overcome this limitation, attention mechanisms have been used in various MIL methodologies [2, 23, 19, 9, 10, 20]. In other deep learning research [32, 35, 49, 50, 51], attention mechanisms have shown significant performance improvements by capturing relationships within the data. Due to these characteristics, attention weights, based on the features of all instances, assign higher weights to important instances for bag predictions, making them useful for instance-level predictions as well [2, 19, 20]. These weights have been shown to improve the prediction performance of instances [23, 9, 10].

### A.2.2 Level of Aggregation

Deep MIL requires aggregating the values of instances to perform predictions for the bag. Early deep MIL models performed individual predictions for instances and then aggregated these predicted values using differentiable pooling operators [5]. However, the approach of making individual predictions for instances was not effective in improving the prediction performance for the bag. To enhance bag prediction performance, new methods were proposed that perform pooling operations on features extracted from each instance to obtain a feature representation for the bag and then learn the bag. These methods apply the classifier used for the bag to instances [5] or cluster the features of instances to find similar instances [6].

In the case of deep MIL utilizing the attention mechanism, early methods enhanced performance by multiplying the features of each instance by attention weights, performing mean pooling to extract a feature representation for the bag, and using that feature to train the bag [2, 19, 20]. This approach suggested that instances with higher attention weights contributed more to the bag's prediction. However, while feature-level aggregation allowed for effective bag predictions, it did not facilitate detailed predictions for individual instances. Conversely, applying attention and aggregating the prediction results for instances achieved high performance [9, 10]. Due to this trend, recent research prefers aggregating the prediction results for instances.

## A.3 Theoretical Study of Multiple Instance Learning

According to Babenko et al. [14], early theoretical research on MIL focused on simple algorithms such as axis-aligned rectangles, which facilitated the derivation of theoretical results [11, 12]. Blum et al. [13] proposed an algorithm that illustrated the relationship between supervised learning and MIL, showing that if a supervised learning algorithm can learn a concept space, MIL can also learn within that space. Sabato et al. [17] addressed the limitations of previous MIL research, which often derived theoretical results for specific domains or simple hypothesis classes, by presenting a generalized theoretical framework applicable to various hypothesis classes and complex data structures. This framework allowed the evaluation of learnability for bags without being limited to specific algorithms. Raff et al. [18] argued that deep MIL algorithms do not adhere to standard MIL assumptions and are unsuitable for real-world environments. They proposed an algorithm unit test to verify adherence to these assumptions, ensuring that models can accurately predict bags in real-world scenarios.

However, all theoretical research proposed on MIL thus far has focused solely on whether MIL can learn bags as effectively as supervised learning without addressing the primary purpose of MIL-predicting individual instances. Therefore, to address this issue, this study proposes a theoretical framework that can universally evaluate whether an algorithm is learnable at the instance level.

# B Notations

The key notations used in Definitions, Theorems, Conditions, and Proofs in this study are summarized in Table 5.

Table 5: Notations.

| Notations | Description |
| --- | --- |
| **Feature Spaces** | |
| $\mathcal{X}$ | bag-level feature space, $\mathcal{X} := \{\mathcal{X}_{inst_1}, \mathcal{X}_{inst_2}, ..., \mathcal{X}_{inst_{N-1}}, \mathcal{X}_{inst_N}\}$ |
| $\mathcal{X}_{inst_i}$ | the feature of the $i^{th}$ instance in the bag |
| **label Spaces** | |
| $\mathcal{Y}$ | bag label space $\mathcal{Y} := \{1, ...k\}$ |
| $\mathcal{Y}_{inst_i}$ | instance label space $\mathcal{Y} := \{1, ...k\}$ |
| **Domains** | |
| $D_{XY}$ | joint distribution on $\mathcal{X} \times \mathcal{Y}$, $D_{XY} := \bigcup_{i=1}^{N} D_{X_{inst_i}Y}$ |
| $D_{X_{inst_i}Y}$ | joint distribution on $\mathcal{X}_{inst_i} \times \mathcal{Y}$ |
| **Domain Spaces** | |
| $\mathcal{D}_{XY}$ | the set space composed of $D_{XY}$ |
| $\mathcal{D}_{XY}^{Ind}$ | the bag domain space with mutually independent domains for all instances |
| $\mathcal{D}_{XY}^{Dep}$ | the bag domain space with mutually dependent domains for all instances |
| $\mathcal{D}_{XY}^{Gen}$ | the entire bag domain space, $\mathcal{D}_{XY}^{Ind} \cup \mathcal{D}_{XY}^{Gen}$ |
| **Hypothesis Spaces** | |
| $\mathcal{H}_{bag}$ | bag hypothesis Space |
| $\mathcal{H}_{inst_i}$ | $i^{th}$ instance Hypothesis Space, $\mathcal{H}_{inst_i} = \{h_i : h_i(X_i) \rightarrow Y_i\}$ |
| $\mathcal{H}_{add_i}$ | extra hypothesis space from external values for $i^{th}$ instance |
| **Risks** | |
| $R_{bag}$ | risk corresponding to bag domain $D_{XY}$ |
| $R_{inst_i}$ | risk corresponding to instance domain $D_{X_{inst_i}Y}$ |
| $R_D$ | risk corresponding to domain $D$ |

## C Theoretical Proofs

### C.1 Proof of Theorem 1

**Theorem 1** If MIL algorithm $A$ satisfies Condition 1, then this algorithm is not PAC learnable for any instance domain space $\mathcal{D}_{X_{inst_i} Y}$ and instance hypothesis space $\mathcal{H}_{inst_i} \subset \{h_{inst_i} : X \to Y\}$.

$$\mathbb{P}\left[\bigcup_{i=1}^{n} |R_{inst_i}(A(S_{inst_i})) - \inf_{h \in \mathcal{H}_{inst_i}} R_{inst_i}(h)| > \epsilon\right] > \delta$$

**Condition 1** The MIL algorithm $A$ is not PAC learnable for the given domain space $\mathcal{D}_{XY}$ and bag hypothesis space $\mathcal{H}_{bag} \subset \{h_{bag} : X \to Y\}$:

$$\mathbb{P}\left[|R_{bag}(A(S)) - \inf_{h \in \mathcal{H}_{bag}} R_{bag}(h)| > \epsilon\right] > \delta$$

**Proof** The performance of MIL algorithms on bags directly depends on the performance of the instances composing those bags. Therefore, the probability that a bag's performance does not reach the optimal hypothesis is greater than or equal to the probability that each individual instance does not reach the optimal hypothesis:

$$\mathbb{P}\left[\bigcup_{i=1}^{n} |R_{inst_i}(A(S_{inst_i})) - \inf_{h \in \mathcal{H}_{inst_i}} R_{inst_i}(h)| > \epsilon\right] \geq \mathbb{P}\left[|R_{bag}(A(S)) - \inf_{h \in \mathcal{H}_{bag}} R_{bag}(h)| > \epsilon\right]$$

If Condition 1 is satisfied, the probability that the bag's performance does not reach the optimal hypothesis can be expressed as follows:

$$\mathbb{P}\left[|R_{bag}(A(S)) - \inf_{h \in \mathcal{H}_{bag}} R_{bag}(h)| > \epsilon\right] > \delta$$

Ultimately, if the bag is not learnable, then one or more instances become unlearnable.

$$\mathbb{P}\left[\bigcup_{i=1}^{n} |R_{inst_i}(A(S_{inst_i})) - \inf_{h \in \mathcal{H}_{inst_i}} R_{inst_i}(h)| > \epsilon\right] > \delta$$

### C.2 Proof of Theorem 2

**Theorem 2** If a MIL algorithm satisfies Condition 4 in $\mathcal{D}_{XY}^{Ind}$, it is learnable for instances.

**Condition 4** The optimal hypothesis for $\mathcal{D}_{XY}^{Ind}$ must be equal to the sum of optimal hypotheses for each individual instance space within $\mathcal{D}_{XY}^{Ind}$:

$$\inf_{h \in \mathcal{H}} R_{\mathcal{D}_{XY}^{Ind}} = \sum_{i=1}^{N} \inf R_{inst_i}$$

**Proof** First, let's assume that the MIL algorithm satisfies Condition 4 in $\mathcal{D}_{XY}^{Ind}$. Based on this assumption, we apply Definition 4 to verify the learnability of MIL in the independent bag hypothesis space.

The learnability in the independent bag hypothesis space means that the algorithm is likely to return predictions within an acceptable error margin from the true optimal hypothesis:

$$\mathbb{P}_{S \sim D_{XY}^m}\left[|R_D(A(S)) - \inf_{h \in \mathcal{H}_{bag}} R_D(h)| \leq \epsilon\right] \geq 1 - \delta$$

Here, $S$ represents the training data, and $A(S)$ denotes the predictions returned by the MIL algorithm. $R_D(h)$ represents the actual risk of hypothesis $h$. Now, this probability is expanded as follows:

$$\mathbb{P}_{S \sim D_{XY}^m} \left[ |R_D(A(S)) - \inf_{h \in \mathcal{H}_{bag}} R_D(h)| \leq \epsilon \right] = \mathbb{P}_{S \sim D_{XY}^m} \left[ R_D(A(S)) \leq \inf_{h \in \mathcal{H}_{bag}} R_D(h) + \epsilon \right]$$

Now, this probability can be expressed as follows:

$$\mathbb{P}_{S \sim D_{XY}^m} \left[ R_D(A(S)) \leq \inf_{h \in \mathcal{H}_{bag}} R_D(h) + \epsilon \right] = \sum_{i=1}^{N} \mathbb{P}_{S \sim D_{X_{inst_i} Y}^m} \left[ R_D(A(S)) \leq \inf_{h \in \mathcal{H}_{inst_i}} R_D(h) + \epsilon \right]$$

In the above equation, $N$ represents the number of instances.

This equation implies that the learnability of bags and instances becomes equivalent. Since we assumed that bags are learnable according to Assumption 1, Condition 4 becomes a necessary and sufficient condition for the learnability of instances.

## C.3  Proof of Theorem 3

**Theorem 3** If a MIL algorithm satisfies Condition 7 in $\mathcal{D}_{XY}^{Gen}$, it is learnable for instances.

**Condition 7** The optimal hypothesis for $\mathcal{D}_{XY}^{Gen}$ must guarantee that it is equal to the weighted sum of optimal hypotheses for each individual instance space within $\mathcal{D}_{XY}^{Gen}$:

$$\inf_{h \in \mathcal{H}} R_{\mathcal{D}_{XY}^{Gen}} = \sum_{i=1}^{N} \alpha_i \inf R_{inst_i} \quad \text{s.t.} \quad \sum_{i=1}^{N} \alpha_i = 1, \quad 0 \leq \alpha_i \leq 1$$

**Proof** To demonstrate that the learnability of bags is equivalent to the learnability of instances, we need to show that the following conditions are satisfied:

$$\mathbb{P} \left[ \left| \inf R_{D_{XY}^{Gen}} - \sum_{i=1}^{N} \alpha_i \inf R_{inst_i} \right| > \epsilon \right] \leq \delta$$

This equation represents that the weighted sum of risks for bags and instances exceeds the permissible error $\epsilon$ with probability $\delta$ or less. Therefore, if this equation is satisfied, the learnability of bags and instances becomes equivalent.

In other words, it indicates that the probability that the difference between the risk for each instance and the weighted sum of the entire bag risks is within the permissible error $\frac{\epsilon}{N}$ is at least $1 - \delta$.

$$\mathbb{P} \left[ \forall i, \left| \inf R_{D_{XY}^{Gen}} - \alpha_i \inf R_{inst_i} \right| \leq \frac{\epsilon}{N} \right] \geq 1 - \delta$$

Through the inequality for the risk of each instance, we can obtain the following:

$$\left| \alpha_i \inf R_{inst_i} - \inf R_{D_{XY}^{Gen}} \right| \leq \frac{\epsilon}{N}, \quad \forall i$$

To prove this, let's define the random variable $X_{inst_i}$ as the difference in risk for each instance:

$$X_{inst_i} = \inf R_{D_{XY}^{Gen}} - \alpha_i \inf R_{inst_i}$$

Let's set the range of each $X_{inst_i}$ to $[-M, M]$, and define $c_i$ as the mean of $X_{inst_i}$. $c_i$ becomes 0 due to cancellation between $\mathbb{E}[\inf R_{D_{XY}^{Gen}}]$ and $\mathbb{E}[\alpha_i \inf R_{inst_i}]$.

$$c_i = \mathbb{E}[X_{inst_i}] = 0$$

Let's define the function $f(X_{inst_1}, X_{inst_2}, \ldots, X_{inst_N})$ as the optimal risk over the general bag domain $D_{XY}^{Gen}$:

$$f(X_{inst_1}, X_{inst_2}, \ldots, X_{inst_N}) = \inf R_{D_{XY}^{Gen}}$$

In this setup, the instances within the bag are not independent; instead, they interact and contribute collectively to the overall bag risk. To capture the effect of each instance on the bag's optimal risk, we introduce weights $\alpha_i$ that represent the contribution of each instance's risk to the total risk. This allows us to define the expected value of $f$ as the weighted sum of the optimal risks of each instance domain:

$$\mathbb{E}[f(X_{inst_1}, \ldots, X_{inst_N})] = \sum_{i=1}^{N} \alpha_i \inf R_{inst_i}$$

This expression reflects the dependencies and interactions between instances, allowing us to represent the general bag risk as a function of each instance's weighted risk.

In the general bag domain space, where one or more instances may have dependent relationships, we apply the Azuma-Hoeffding inequality.

$$\mathbb{P}\left[|f(X_{inst_1}, \ldots, X_{inst_N}) - \mathbb{E}[f(X_{inst_1}, \ldots, X_{inst_N})]| > t\right] \leq 2\exp\left(\frac{-2t^2}{\sum_{i=1}^{N} c_i^2}\right)$$

Since $f(X_{inst_1}, X_{inst_2}, \ldots, X_{inst_N})$ represents the risk of the bag as stated earlier, the equation can be written as follows:

$$\mathbb{P}\left[|\inf R_{D_{XY}^{Gen}} - \mathbb{E}[\inf R_{D_{XY}^{Gen}}]| > t\right] \leq 2\exp\left(\frac{-2t^2}{\sum_{i=1}^{N} c_i^2}\right)$$

The $t$ term in $|\inf R_{D_{XY}^{Gen}} - \mathbb{E}[\inf R_{D_{XY}^{Gen}}]|$ represents the deviation from the expected value, so it can be replaced by the permissible error $\epsilon$ for learnability, yielding the following:

$$\mathbb{P}\left[|\inf R_{D_{XY}^{Gen}} - \mathbb{E}[\inf R_{D_{XY}^{Gen}}]| > \epsilon\right] \leq 2\exp\left(\frac{-2\epsilon^2}{\sum_{i=1}^{N} c_i^2}\right)$$

Finally, it can be expressed as follows:

$$\mathbb{P}\left[\left|\inf R_{D_{XY}^{Gen}} - \sum_{i=1}^{N} \alpha_i \inf R_{inst_i}\right| > \epsilon\right] \leq 2\exp\left(\frac{-2\epsilon^2}{\sum_{i=1}^{N} c_i^2}\right)$$

$c_i$ represents the mean of errors. Since Deep MIL algorithms are trained to minimize the mean error, $\sum_{i=1}^{N} c_i \simeq 0$.

$$\mathbb{P}\left[\left|\inf R_{bag} - \sum_{i=1}^{N} \alpha_i \inf R_{inst_i}\right| > \epsilon\right] \leq \delta$$

This ensures that the sum of $c_i$ values remains within a certain range, guaranteeing that it is smaller than the probability of being trained within the permissible error range $\epsilon$, denoted as $\delta$.

$$2\exp\left(\frac{-2\epsilon^2}{\sum_{i=1}^{N}c_i^2}\right) \leq \delta \Leftrightarrow \frac{1}{\sum_{i=1}^{N}c_i^2} \geq \frac{1}{2\epsilon^2}\log\frac{2}{\delta}$$

Therefore, under the assumption of successful learning, Condition 2 is satisfied. Thus, a well-trained MIL model becomes learnable for instances.

### C.4 Proof of Theorem 4

**Theorem 4** In $\mathcal{D}_{XY}^{Ind}$, the MIL algorithm is PAC learnable for bags.

**Proof** The PAC learnability of $\mathcal{D}_{XY}^{Ind}$ implies that every concept space $\mathcal{C}_{\mathcal{D}_{XY}^{Ind}}$ can be approximated by hypothesis spaces $\mathcal{H}_{D_{XY}^{Ind}}$.

$$\mathcal{C}_{\mathcal{D}_{XY}^{Ind}} \subseteq \mathcal{H}_{D_{XY}^{Ind}}$$

Since $\mathcal{D}_{XY}^{Ind}$ assumes that predictions for all instances are independent, predictions for bags become the union of predictions for instances.

$$\mathcal{H}_{D_{XY}^{Ind}} = \bigcup_{i=1}^{n} \mathcal{H}_{inst_i}$$

The hypothesis space $\mathcal{H}_{bag_{non-att}}$ for Non-Attention MIL algorithms, which do not use Attention, is also the union of predictions of instances:

$$\mathcal{H}_{bag_{non-att}} = \bigcup_{i=1}^{n} \mathcal{H}_{inst_i}$$

Thus, $\mathcal{H}_{bag_{non-att}}$ becomes equivalent to the hypothesis space $\mathcal{H}_{D_{XY}^{Ind}}$ of PAC Learnable $\mathcal{D}_{XY}^{Ind}$:

$$\mathcal{H}_{D_{XY}^{Ind}} = \mathcal{H}_{bag_{non-att}}$$

Therefore, in the case of PAC Learnable scenarios, Non-Attention MIL is learnable.

The hypothesis space $\mathcal{H}_{bag_{att}}$ of Attention MIL encompasses the hypothesis space $\mathcal{H}_{bag_{non-att}}$ of Non-Attention MIL:

$$\mathcal{H}_{bag_{non-att}} \subseteq \mathcal{H}_{bag_{att}}$$

Thus, if $\mathcal{D}_{XY}^{Ind}$ is PAC Learnable, Attention MIL is also learnable.

Ultimately, if $\mathcal{D}_{XY}^{Ind}$ is PAC Learnable, then all MIL algorithms can learn from bags.

### C.5 Proof of Theorem 5

**Theorem 5** In $\mathcal{D}_{XY}^{Gen}$, MIL algorithms that aggregate independent hypothesis spaces for each instance must utilize attention satisfying **Condition 8** to be PAC learnable for bags.

**Condition 8** The hypothesis space $\mathcal{H}_{bag_{att}}$ of Attention MIL should be equal to the sum of independent hypothesis spaces $h_{inst_i}$ multiplied by attention weights $Att_i$:

$$\mathcal{H}_{bag_{att}} = \left\{ h_{bag_{att}} \mid h_{bag_{att}} = \sum_{i=1}^{n} Att_i \cdot h_{inst_i}, \text{ where } 0 < Att_i < 1, \sum_{i=1}^{n} Att_i = 1 \right\}$$

**Proof** The PAC Learnability of $\mathcal{D}_{XY}^{Gen}$ implies that all concept spaces $\mathcal{C}_{\mathcal{D}_{XY}^{Gen}}$ of $\mathcal{D}_{XY}^{Gen}$ can be approximated by the hypothesis space $\mathcal{H}_{D_{XY}^{Gen}}$ of $\mathcal{D}_{XY}^{Gen}$:

$$\forall \mathcal{H}_{D_{XY}^{Gen}} \in \mathcal{C}_{\mathcal{D}_{XY}^{Gen}}$$

$\mathcal{H}_{D_{XY}^{Gen}}$ represents the union space of hypotheses $\mathcal{H}_{inst_i}^{Gen}$ with dependencies within the same bag. The optimal hypothesis within each instance space can be expressed as:

$$\inf R_{inst_i}^{Gen} = \inf_{h_{inst_i}^{Gen} \in \mathcal{H}_{inst_i}^{Gen}} R_{inst_i}^{Gen}$$

Combining these optimal hypotheses yields:

$$\bigcup_{i=1}^{n} \inf R_{inst_i}^{Gen} = \sum_{i=1}^{n} \inf R_{inst_i}^{Gen}$$

From previous proofs, it's shown that the Azuma-Hoeffding inequality satisfies:

$$\mathbb{P}\left[\left|\inf R_{D_{XY}^{Gen}} - \sum_{i=1}^{N} \alpha_i \inf R_{inst_i}^{Gen}\right| > \epsilon\right] \leq \delta$$

Thus, the hypothesis space $\mathcal{H}_{bag}^{Gen}$ for $\mathcal{D}_{XY}^{Gen}$ can be expressed as:

$$\mathcal{H}_{D_{XY}^{Gen}} = \left\{ h_{D_{XY}^{Gen}} \mid h_{D_{XY}^{Gen}} = \sum_{i=1}^{n} \alpha_i \cdot h_{inst_i}, \text{ where } 0 < \alpha_i < 1, \sum_{i=1}^{n} \alpha_i = 1 \right\}$$

In proving the learnability of bags for $\mathcal{D}_{XY}^{Gen}$, it's shown that the hypothesis space for Non-Attention MIL algorithms is:

$$\mathcal{H}_{bag_{non-att}} = \bigcup_{i=1}^{n} \mathcal{H}_{inst_i}$$

The hypothesis space $\mathcal{H}_{D_{XY}^{Gen}}$ extends beyond the range of hypotheses that can independently predict all instances, incorporating dependencies with weighted hypotheses $\alpha_i$:

$$\mathcal{H}_{bag_{non-att}} \subseteq \mathcal{H}_{D_{XY}^{Gen}}$$

$$\mathcal{H}_{D_{XY}^{Gen}} \not\subseteq \mathcal{H}_{bag_{non-att}}$$

Thus, Non-Attention MIL Algorithms are not learnable from $\mathcal{D}_{XY}^{Gen}$.

However, according to Condition 8, the hypothesis space of Attention MIL Algorithms is:

$$\mathcal{H}_{bag_{att}} = \left\{ h_{bag_{att}} \mid h_{bag_{att}} = \sum_{i=1}^{n} Att_i \cdot h_{inst_i}, \text{ where } 0 < Att_i < 1, \sum_{i=1}^{n} Att_i = 1 \right\}$$

Here, replacing $Att_i$ with $\alpha_i$ makes $\mathcal{H}_{bag_{att}}$ identical to $\mathcal{H}_{D_{XY}^{Gen}}$:

$$\mathcal{H}_{bag_{att}} = \mathcal{H}_{D_{XY}^{Gen}}$$

Thus, Attention MIL is learnable from $\mathcal{D}_{XY}^{Gen}$ in terms of bags.

## C.6 Proof of Lemma 1

**Lemma 1** Condition 9 serves as a necessary condition for the learnability of instances, when the hypothesis space for the $i^{th}$ instance of a MIL algorithm is $\mathcal{H}_{inst_i} \cup \mathcal{H}_{add_i}$. Here, $\mathcal{H}_{inst_i}$ represents the hypothesis space for the $i^{th}$ instance, and $\mathcal{H}_{add_i}$ denotes the hypothesis space for the $i^{th}$ instance generated through elements outside the $i^{th}$ instance.

**Condition 9** $\mathcal{H}_{add_i}$ must be a subset of $\mathcal{H}_{inst_i}$:

$$\mathcal{H}_{inst_i} \supset \mathcal{H}_{add_i} := \{h_{add_i} : \mathcal{X}_{add_i} \to \mathcal{Y}\}$$

**Proof**

When $\mathcal{H}_{inst_i} \not\subset \mathcal{H}_{add_i} := \{h_{add_i} : \mathcal{X}_{add_i} \to \mathcal{Y}\}$, the hypothesis space $\mathcal{H}_{bag}$ for bags is as follows:

$$\mathcal{H}_{bag} = \bigcup_{i=1}^{N} \mathcal{H}_{inst_i} \cup \mathcal{H}_{add_i}$$

In this case, using the Azuma-Hoeffding Inequality to compare the deviation between $R_{bag}$ and each $R_{inst_i}$, we get:

$$\mathbb{P}\left( \left| \frac{1}{N} \sum_{i=1}^{N} R_{inst_i} - R_{bag} \right| \geq \epsilon \right) \leq \frac{\sum_{i=1}^{N} \text{Var}[R_{inst_i}]}{N^2 \epsilon^2}$$

If $h_{add_i}$ is not a subset of $h_{inst_i}$, additional hypothesis space exists, resulting in a deviation between the risk of the Bag-level hypothesis and the risks of each Instance.

Hence, the deviation between $R_{bag}$ and $R_{inst_i}$ is greater than or equal to 0:

$$\inf_{h \in \mathcal{H}} R_{bag} \leq \sum_{i=1}^{N} \inf R_{inst_i}$$

Therefore, it fails to satisfy the necessary and sufficient condition for the learnability of instances in $D_{XY}^{Ind}$. The independent bag domain space implies that all weights in the general bag domain space are equal, thus the relation is as follows:

$$D_{XY}^{Ind} \subset D_{XY}^{Gen}$$

Hence, if it's not learnable in the independent bag domain space, it means it's not learnable in the general bag domain space. Therefore, through the proposed theoretical framework, Condition 7 becomes a necessary condition for MIL.

Thus, Condition 9 becomes a necessary condition for learning instances in MIL.

## C.7 Proof of Theorem 6

**Theorem 6** In $\mathcal{D}_{XY}^{Ind}$, MIL algorithms that perform instance-pooling are PAC learnable for instances.

**Proof**

First, the learnability of MIL algorithms on instances in $\mathcal{D}_{XY}^{Ind}$ means the algorithm satisfies Condition 4:

$$\inf_{h \in \mathcal{H}} R_{\mathcal{D}_{XY}^{Ind}} = \sum_{i=1}^{N} \inf R_{inst_i}$$

Performing Instance-Pooling means forming bag-level hypotheses by independently combining hypotheses at each instance level. The error at the bag level derived from instance-level pooling can be defined as follows:

$$R_{\mathcal{D}_{XY}^{Ind}}(h_{bag}) = \mathbb{E}_{S \sim \mathcal{D}_{XY}^{Ind}}[\ell(h_{bag}(X), Y)]$$

We can express the error at each instance level as:

$$\inf R_{inst_i} = \inf_{h_i \in \mathcal{H}_{inst_i}} R_{\mathcal{D}_{X_{inst_i}Y}^{Ind}}$$

Now, comparing the two errors, we have:

$$
\begin{aligned}
\inf_{h \in \mathcal{H}} R_{\mathcal{D}_{XY}^{Ind}}(h) &= \inf_{h \in \mathcal{H}} \mathbb{E}_{S \sim \mathcal{D}_{XY}^{Ind}}[\ell(h(X), Y)] \\
&= \mathbb{E}_{(X,Y) \sim \mathcal{D}_{XY}^{Ind}} \left[ \inf_{h \in \mathcal{H}} \ell(h(X), Y) \right] \\
&= \mathbb{E}_{(X,Y) \sim \mathcal{D}_{XY}^{Ind}} \left[ \sum_{i=1}^{N} \inf_{h_i \in \mathcal{H}_{inst_i}} \ell(h_i(X_{inst_i}), Y_{inst_i}) \right] \\
&= \sum_{i=1}^{N} \mathbb{E}_{S \sim \mathcal{D}_{X_{inst_i}Y}^{Ind}} \left[ \inf_{h_i \in \mathcal{H}_{inst_i}} \ell(h_i(X_{inst_i}), Y_{inst_i}) \right] \\
&= \sum_{i=1}^{N} \inf_{h_i \in \mathcal{H}_{inst_i}} \mathbb{E}_{(X_{inst_i}, Y_{inst_i}) \sim \mathcal{D}_{X_{inst_i}Y}^{Ind}} [\ell(h_i(X_{inst_i}), Y_{inst_i})] \\
&= \sum_{i=1}^{N} \inf_{h_i \in \mathcal{H}_{inst_i}} R_{\mathcal{D}_{X_{inst_i}Y}^{Ind}}
\end{aligned}
$$

Thus, performing pooling on instance predictions satisfies Condition 4.

This proves that MIL algorithms can learn from instances in $\mathcal{D}_{XY}^{Ind}$.

## C.8 Proof of Theorem 7

**Theorem 7** MIL algorithms that perform Embedding-Pooling are not learnable for instances.

**Proof**

In Embedding-pooling, features of each instance are combined using an aggregation function $g$:

$$F(X) = g(f_1(X_1), f_2(X_2), \ldots, f_n(X_n))$$

Here, $f_i(X_i)$ represents the feature of each $i^{th}$ instance, and $g$ is a function that integrates features to generate a single vector. Consequently, $\mathcal{H}_{add_i}$ includes hypotheses based on $F(X)$, making $\mathcal{H}_{add_i}$ dependent on the features of all instances, $f_1(X_1), f_2(X_2), \ldots, f_n(X_n)$:

$$\mathcal{H}_{add_i} = \{h : F(X) \to Y\}$$

On the other hand, $\mathcal{H}_{inst_i}$ produces results dependent solely on the $i^{th}$ instance feature:

$$\mathcal{H}_{inst_i} = \{h_i : h_i(X_i) \to Y_i\}$$

A function $h \in \mathcal{H}_{add_i}$ using $F(X)$ cannot belong to $\mathcal{H}_{inst_i}$. This is because functions in $\mathcal{H}_{inst_i}$ only use $X_i$ as input, while $h$ is based on $F(X)$, i.e., the combined result of all $X_i$ features. Since

functions in $\mathcal{H}_{add_i}$ have more complex dependencies beyond the scope of $\mathcal{H}_{inst_i}$, the following inequality holds:

$$\mathcal{H}_{inst_i} \not\supset \mathcal{H}_{add_i}$$

Therefore, MIL algorithms using Embedding-pooling fail to satisfy Condition 9, rendering them incapable of learning from instances.

## C.9 Proof of Theorem 8

**Theorem 8** If the MIL algorithm does not adhere to Condition 10, it is not learnable for instances.

**Condition 10** The risk $R_{inst_i}$ for the $i^{th}$ instance should be as follows:

$$R_{inst_i} = \mathbb{E}_{(x_{inst_i},y) \sim D_{X_{inst_i}Y}} \ell_{inst_i}(h, y) \quad , \text{where} \quad h \in \mathcal{H}_{inst_i} \cup \mathcal{H}_{bag-level_i}$$

**Proof**

If $h_{bag-level}$ is not empty, then the following holds:

$$\forall \delta > 0, \exists h \in \mathcal{H}_{bag} : R_{bag}(h) > \sum_{i=1}^{N} R_{inst_i}(h_{inst_i}) + \delta$$

For every $\delta > 0$, there exists at least one predictor belonging to $h_{bag-level}$ that can make the lower bound of $R_{bag}$ greater than the sum of the lower bounds of $\sum_{i=1}^{N} R_{inst_i}$. This implies that if $h_{bag-level}$ is not empty, it fails to satisfy Condition 9. Thus, $h_{bag-level}$ must be empty for Condition 4 to hold.

As a result, Condition 10 also becomes a necessary condition for MIL algorithms to be learnable from instances.

## C.10 Proof of Theorem 9

**Theorem 9** When MIL algorithms use conjunctive pooling for aggregation in $\mathcal{D}_{XY}^{Gen}$, they are learnable for instances.

**Proof**

Condition 7 being satisfied implies learnability from instances:

$$\inf_{h \in \mathcal{H}} R_{\mathcal{D}_{XY}^{Gen}} = \sum_{i=1}^{N} \alpha_i \inf R_{inst_i}$$

Here, $\alpha_i$ represents the weight for the $i^{th}$ instance, satisfying the following conditions:

$$\alpha_i \in (0, 1), \sum_{i=1}^{N} \alpha_i = 1$$

Given Assumption 1 where the MIL algorithm is assumed to be PAC learnable for bags, $R_{\text{bag}}$ becomes the lower bound of prediction error for bags, and $R_{\text{inst}_i}$ becomes the lower bound of prediction error for each instance. Thus, Condition 7 can be expressed as:

$$\inf_{h \in \mathcal{H}} R_{\text{bag}} = \sum_{i=1}^{N} \alpha_i \inf R_{\text{inst}_i}$$

For Conjunctive Pooling, the optimal hypothesis $h_{\text{bag}}$ for bags is:

$$h_{\text{bag}} = \sum_{i=1}^{N} \alpha_i h_{\text{inst}_i}$$

The lower bound for the optimal hypothesis $h_{\text{inst}_i}$ for the $i^{th}$ instance is:

$$\inf R_{\text{inst}_i} = R(h_{\text{inst}_i})$$

Therefore, the lower bound of prediction error for the optimal hypothesis $h_{\text{bag}}$ for bags is:

$$
\begin{aligned}
\inf_{h \in \mathcal{H}} R_{\text{bag}} &= R(h_{\text{bag}}) \\
&= R\left( \sum_{i=1}^{N} \alpha_i h_{\text{inst}_i} \right) \\
&= \inf_{h \in \mathcal{H}} \mathbb{E}_{(x,y) \sim \mathcal{D}_{XY}^{Gen}} \left[ \ell\left( \sum_{i=1}^{N} \alpha_i h_{\text{inst}_i}(x), y \right) \right] \\
&= \sum_{i=1}^{N} \alpha_i \inf R_{\text{inst}_i}
\end{aligned}
$$

Ultimately, MIL algorithms using Conjunctive Pooling satisfy Condition 7 and are learnable from instances.

## C.11   Proof of Theorem 10

**Theorem 10** If the MIL algorithm extracts features of instances through RNN-based neural networks for aggregation, it is unable to learn from instances.

**Proof** When extracting features through an RNN, the hypothesis space of the $i^{th}$ instance's features includes not only the hypothesis spaces based on the $i^{th}$ instance's information but also those based on the information of preceding instances up to the $i^{th}$ one:

$$\mathcal{H}_{inst_i - rnn} = \bigcup_{j=1}^{i} \mathcal{H}_{inst_j}$$

Therefore, when extracting features of instances through neural networks like RNNs, if $\mathcal{H}_{inst_i} := \{h_{inst_i} : X_{inst_i} \to Y\}$, then $\mathcal{H}_{bag-level}(x) := \{h_{bag-level} : \mathcal{X} \to \mathcal{Y}\} \neq \emptyset$:

$$
\begin{aligned}
R_{inst_i} &= \mathbb{E}_{(x_{inst_i}, y) \sim D_{X_{inst_i} Y}} \ell_{inst_i}(h, y) \quad \text{where} \quad h \in \mathcal{H}_{inst_i} \cup \mathcal{H}_{bag-level} \\
&= \mathbb{E}_{(x_{inst_i}, y) \sim D_{X_{inst_i} Y}} \ell_{inst_i}(h, y) \quad \text{where} \quad h \in \mathcal{H}_{inst_i} \cup \mathcal{H}_{inst_i - rnn}, \mathcal{H}_{inst_i - rnn} \neq \emptyset
\end{aligned}
$$

This means that when performing MIL using features extracted through RNNs, it fails to satisfy Condition 10, a necessary condition for learning from instances. Hence, it becomes incapable of learning from instances.

Therefore, it fails to satisfy Condition 10.

As a side note, in the case of extracting features through bidirectional RNNs, the size of the hypothesis space would be:

$$\mathcal{H}_{inst_i - birnn} = \bigcup_{j=1}^{N} \mathcal{H}_{inst_j}$$

So, the same conclusion applies.

## D  Experimental Setting

### D.1  Experimental Validation of Theorem 5

Theorem 5 explains that Instance pooling MIL, which aggregates instance hypothesis spaces independently, does not satisfy Condition 8 and thus is not learnable with respect to $D_{XY}^{Gen}$. In contrast, the Conjunctive Pooling model, which multiplies weights by the hypothesis space of instances to reflect the relationships among them, is learnable.

To validate this, we conduct comparative experiments on the following synthetic datasets, assuming $D_{XY}^{Gen}$ outside of $D_{XY}^{Ind}$. Each bag in the dataset is labeled based on the relationships between the ten instances constituting the bag, and one of four labels is assigned accordingly.

- **Label 1**: If the bag contains both 3 and 5, the bag's label is 1.
- **Label 2**: If the bag contains 1 but not 7, the bag's label is 2.
- **Label 3**: If the bag contains 1 and also contains 7, the bag's label is 3.
- **Label 0**: Any bag that does not meet the criteria for other labels is assigned the negative label 0.

Under these labeling assumptions, the MNIST dataset's training data is split into training and testing datasets in an 8:2 ratio. Then, ten images are randomly selected to form a bag, and labeling is performed according to the assumptions.

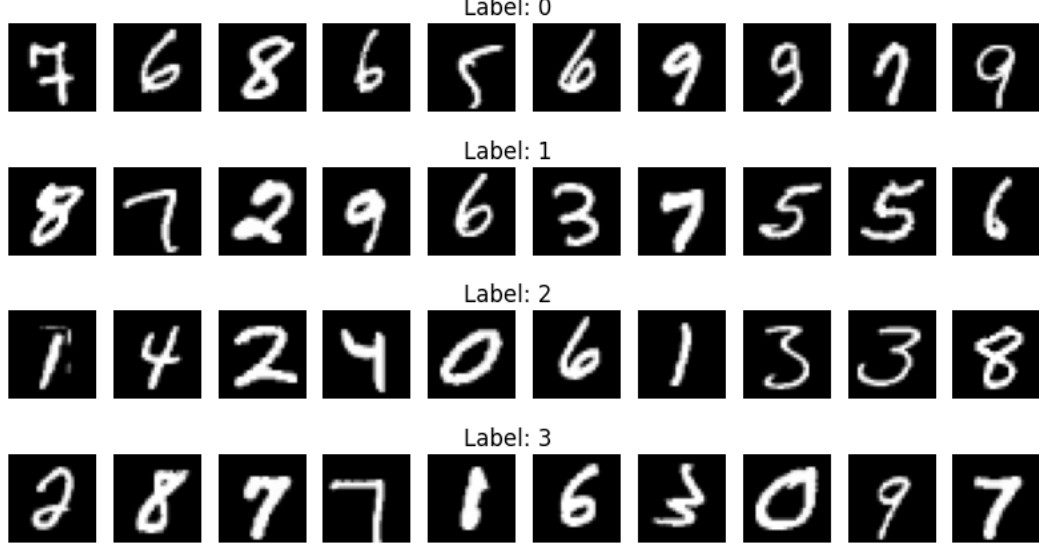

Figure 3: Examples of synthetic dataset to verify Theorem 5.

To extract features of the synthetic dataset, which assumes each MNIST image as an instance, all models use the feature extractor structure described in Table 6. First, mi-Net [5], Attention MIL [2], Additive MIL [9], and Conjunctive MIL [10] performed Aggregation through basic Instance, Attention, Additive, and Conjunctive-pooling operations, respectively. We used the same classifier structure used to perform the pooling operations described in Appendix A.2. The employed attention mechanism is the gated attention mechanism, which is commonly used in MIL research [2]. Second, Causal MIL [21], MIREL [38], Loss-Attention [39], SA-AbMILP [40], and TransMIL [19] performed additional tuning on pooling for performance improvement. We implemented them using the official code provided by each respective paper[2][3][4][5][6]. Third, for MIREL [38], although the prediction modules

---

[2] https://github.com/WeijiaZhang24/CausalMIL?utm_source=catalyzex.com
[3] https://github.com/liupei101/MIREL
[4] https://github.com/xsshi2015/Loss-Attention
[5] https://github.com/gmum/Kernel_SA-AbMILP
[6] https://github.com/szc19990412/TransMIL

for the bag and the instances operate separately, the validation is adjusted to fit the nature of MIL. Specifically, the bag-level prediction module is not used, and instead, the pooled instance predictions are used for bag-level evaluation. Fourth, for Conjunctive MIL [10] and TransMIL [19], while each original paper utilizes positional information that affects the predictions, this factor is excluded from our experiments, as it does not influence the outcomes in our setup.

The operational processes and theoretical validation for Causal MIL [21], MIREL [38], Loss-Attention [39], SA-AbMILP [40], and TransMIL [19] are performed in Appendix F.

Table 6: The architecture of the instance feature extractor.

| Layer | Type |
|---|---|
| 1 | conv(3,1,1)-32 + ReLU |
| 2 | maxpool(2,2) |
| 3 | conv(3,1,1)-64 + ReLU |
| 4 | maxpool(2,2) |
| 5 | flatten |
| 6 | fc-128 + ReLU |

The hyperparameter settings for the models used in the experiments are shown in Table D.

Table 7: Hyper parameter setting.

| Optimizer | Learning rate | Cost funtion | Epochs |
|---|---|---|---|
| Adam | 0.0001 | NLL loss function | 20 |

Since the labels for the bags are multi-class, we used macro-F1 score (Macro-F1), micro-F1 score (Micro-F1), and weighted-F1 score to measure multi-class classification performance. To ensure generalized results, each algorithm was trained ten times with different initializations, and the average performance was measured. All experiments conducted in this study were performed on an Intel(R) Xeon(R) Silver 4210R 40 Core CPU @ 2.40 GHz, 32GB RAM, and NVIDIA RTX A5000.

## D.2 Experimental Validation of Theorems 8 and 9

Theorem 8 and Theorem 9 demonstrate that Attention MIL [2] and Additive MIL [9], which perform attention multiplication on instance features, are not capable of learning at the instance level, whereas Conjunctive MIL is capable of instance-level learning.

To empirically validate this, it is necessary to compare the prediction performance of a Deep MIL model on bags with its performance on instances. Therefore, experiments should be conducted on datasets that have labels for both bags and instances. In this study, we use the labeling criteria for the synthetic dataset explained in Appendix D.1.

The feature extractor and hyper-parameter settings followed the model structure used in Appendix D.1. The performance measurement method also adhered to the approach specified in Appendix D.1. However, due to the nature of MIL, which involves learning whether a specific instance is positive for each class, we also calculated the average AUROC performance for each class and measured the overall AUROC.

To analyze the impact of the attention mechanism, we adjusted the temperature parameter ($\tau$) used in the attention operations to three values: 0.5, 1, and 2. This process was repeated ten times to calculate the average performance. The average performance of each algorithm for the bag and the average performance difference between the bag and instances are discussed in Section 4.4.3. The performance analysis based on the adjustment of $\tau$ is detailed in Appendix E.1.

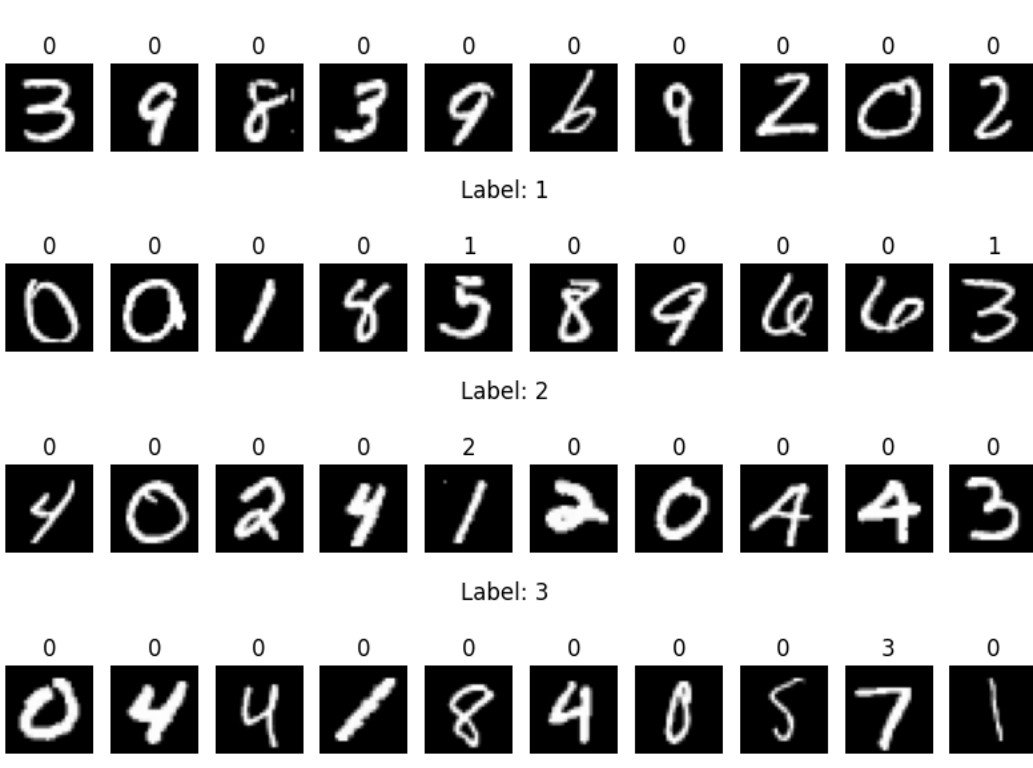

Figure 4: Example synthetic dataset to verify Theorem 8, 9.

### D.3 Experimental Validation of Theorems 10 and 11

Theorem 10 and Theorem 11 demonstrate that using an RNN layer to capture positional dependencies or performing positional encoding can act as factors that hinder the learnability of instances.

To experimentally validate this, we use the WebTraffic dataset [10], which provides label information for each time point in a time-series sliding window data. Both the model structure and hyperparameters are set identical to those in Early et al. [10], and models are configured based on the extent to which features with positional encoding and RNN are used, as following four models[7]:

1. A model that does not use features with positional encoding and RNN for instance prediction and attention operations **(Default)**

2. A model that uses features with positional encoding and RNN for both instance prediction and attention operations **(All)**

3. A model that uses features with positional encoding and RNN only for attention operations **(Att)**

4. A model that uses features with positional encoding and RNN only for instance prediction operations **(Predict)**

All experiments conducted in this study were performed on an Intel(R) Xeon(R) Silver 4210R 40 Core CPU @ 2.40 GHz, 32GB RAM, and NVIDIA RTX A5000.

To evaluate the instance-level prediction performance of deep MIL models, we used the normalized discounted cumulative gain (NDCG@n) and the area under the perturbation curve with random orderings (AOPCR) metrics, as employed by Early et al. [52, 10]. To reliably measure the performance of each model, we repeated the evaluation 10 times and calculated the average performance. AOPCR assesses the interpretability of time-series data by measuring the decrease in model predictions based

---

[7]https://github.com/JAEarly/MILTimeSeriesClassification

on the importance of time points. Starting from the most important time point, the time series is rearranged in order of importance. AOPC is then calculated by removing instances one by one from the most important time points and evaluating how the predictive performance changes.

$$AOPC(\mathbf{X_i}, \mathbf{O_{i,c}}) = \frac{1}{t-1} \sum_{j=1}^{t-1} F_c(\mathbf{X_i}) - F_c(MoRF(\mathbf{X_i}, \mathbf{O_{i,c}}, j)),$$

Here, the $MoRF$ function removes time points in the given order of importance. Instead of removing each time point individually, time points are grouped into blocks corresponding to 5% of the total time series, and perturbations are applied until 50% of the time series is removed, limiting the number of model calls to a maximum of 10. Finally, AOPCR is calculated by comparing the average AOPC value for the given order of importance to the average AOPC value for random orderings to achieve normalization:

$$AOPCR(\mathbf{X_i}, \mathbf{O_{i,c}}) = \frac{1}{T} \sum_{r=1}^{T} \left( AOPC\left(\mathbf{X_i}, \mathbf{O_{i,c}}\right) - AOPC(\mathbf{X_i}, \mathbf{R_i^{(r)}}) \right).$$

Here, $T$ represents the number of repetitions of random orderings. In this study, $T$ is set to 3 following the settings of Early et al. [10].

NDCG@n is a metric used to evaluate instance predictions by comparing them to the ground truth ordering for a specific class. Instances within a bag are classified as supporting instances, neutral instances, or refuting instances based on their labels. The ideal prediction order for instances should be supporting instances, neutral instances, and then refuting instances for the given class. NDCG@n is calculated by ranking the instance predictions $\{\phi_1, \ldots, \phi_k\}$ for the given class from highest to lowest importance and comparing this order to the ground truth order as follows:

$$NDCG@n = \frac{1}{IDCG} \sum_{i=1}^{n} \frac{\text{rel}(i)}{\log_2(i+1)}$$

Here, IDCG is the ideal discounted cumulative gain, which normalizes the score based on the value of $n$, where $n$ is the number of instances with a positive label.

### D.4  Experimental Validation of MD-MIL

In this section, we conduct additional experiments in an extended scenario of Video Anomaly Detection (VAD) to demonstrate that the framework proposed in this study assists in designing optimal algorithms for realistic MD-MIL scenarios. The traditional VAD problem has been addressed as a type of 1D-MIL problem, as shown in Figure 5 (a), where the objective is to detect anomalous snippets based on labels for the entire video. In this context, each snippet should reflect temporal dependencies.

Moreover, the conventional VAD problem can be extended as an MD-MIL problem, as depicted in Figure 5 (b). The problem aims to detect both anomalous snippets and anomalous patches within each snippet based on labels for the entire video. In this case, the patches that constitute a snippet must not only reflect relationships with other patches within the same snippet but, much like multivariate time series data [53, 54], also account for temporal dependencies with corresponding patches in other snippets. Therefore, the VAD problem in an MD-MIL context requires consideration of relationships not only among instances within the same bag but also with instances at the same position across different bags.

To design a model that is capable of learning about instances while reflecting relationships under this complex problem setting, the following requirements must be satisfied:

1. The VAD problem in MD-MIL should reflect relationships among complex multi-dimensional instances. Therefore, this should be addressed by a learnable MIL method (i.e., Conjunctive-pooling) on $D_{XY}^{Gen}$.

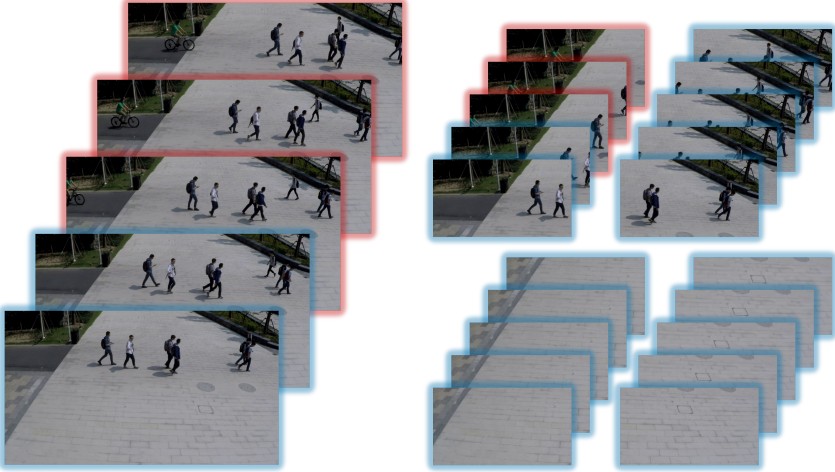

(a) The video data structure consisting of 1-dimensional-instances.

(b) The video data structure consisting of multi-dimensional-instances.

Figure 5: The video data structure consisting of 1-dimensional and multi-dimensional-instances (Blue: Negative, Red: Positive) [55].

2. Additional information about relationships between instances, such as position encoding, should be used in the attention computation process. This ensures learnability about instances and can improve performance, as demonstrated in Section 4.4.3. Based on this, attention operations should incorporate this information to reflect temporal relationships with patches at the same position in other snippets for each patch detection.

To validate these requirements, we measure and compare the performance between the following three types of MD-MIL models:

1. **None-Attention**: This model predicts for higher-dimensional bags through Instance-pooling in each dimension. It independently models the relationships between snippets composing a video and patches composing the same snippet.

2. **Attention**: This model predicts for higher-dimensional Bags by performing Conjunctive-Pooling based on attention weights calculated within each dimension (same bag). Through this structure, it can make predictions reflecting the relationships between snippets composing a video and patches composing the same snippet.

3. **Cross-Attention**: To reflect the temporal dependencies of patches at the same position in different snippet bags, this model computes attention weights based on features of each patch computed through a bidirectional GRU-layer and performs Conjunctive-pooling. Through this structure, it can make predictions reflecting the relationships between snippets composing a video, patches composing the same snippet, and patches at the same position in different snippets.

All models are trained using anomaly labels at the video level. The encoder structure for computing instances is identical across models, as shown in the Table 8, with differences only in pooling and attention operations.

The hyperparameter settings for the models used in the experiments are presented in the following Table 9.

For this study, we utilized the ShanghaiTech dataset [55, 56], a well-established benchmark dataset for Video Anomaly Detection (VAD). As illustrated in Figure 5 (b), each snippet was pre-divided into four patches. To extract features from these patches, we employed a pre-trained I3D-ResNet model [8]. This model was used to extract snippet features for each patch, which were then utilized in our experiments.

---

[8]https://github.com/Tushar-N/pytorch-resnet3d

Table 8: The architecture of the instance feature extractor in the MD-MIL model.

| Layer | Type |
|-------|------|
| 1 | fc-1024 + ReLU |
| 2 | LayerNorm(1024) |
| 3 | fc-1024 + ReLU |
| 4 | LayerNorm(1024) |
| 5 | fc-1024 + ReLU |
| 6 | LayerNorm(1024) |
| 7 | fc-1024 + ReLU |
| 8 | LayerNorm(1024) |

Table 9: Hyper parameter setting.

| Optimizer | Learning rate | Cost funtion | Epochs |
|-----------|---------------|--------------|--------|
| Adam | 0.0001 | NLL loss function | 100 |

# E    Additional Experimental Results

## E.1    Experimental Validation of Theorems 8 and 9

The results of adjusting the variance of attention by applying $\tau$ in the softmax operation to determine how much attention affects the error between bag prediction performance and instance performance in deep MIL are presented in Table 10. While all Attention-pooling MIL models show high performance for bags, they exhibit low learning performance for instances. Conjunctive-pooling MIL, satisfying Theorem 9, shows very little error between bag performance and instance performance, with instances sometimes even outperforming bags. However, in the case of Additive-pooling MIL, as $\tau$ increases and the variance among instances decreases, AUROC between bags and instances decreases. This result suggests that for Additive-pooling MIL to be learnable for instances, the strength of the attention must be weak. In other words, the closer the structure is to instance predict-level pooling, the more likely it is to be learnable for instances.

## E.2    Experimental Validation of MD-MIL

The experimental results indicate that the Cross-Attention method achieved the best performance, followed by the Attention-driven model, while the None-Attention-driven model showed the least favorable performance. These results demonstrate that 1) attention weights are necessary to capture the relationships between video snippets and patches within the same snippet, and 2) when learning relationships between patches in different snippets (i.e., bags) are required, Cross-Attention-driven model, which utilizes such information only during attention computation, is desired.

# F    Interpretation of MIL Algorithms through Our Theoretical Framework

In this section, we demonstrate how the proposed theoretical framework can interpret actual existing MIL algorithms. To further validate our framework's capability to assess instance-level learning in MIL algorithms, we extended our analysis beyond the basic pooling techniques discussed earlier. We additionally selected five MIL algorithms that apply additional tuning to the existing pooling methods for performance improvement. These algorithms were chosen to demonstrate how our framework can generalize to MIL approaches that enhance standard pooling operations through additional optimization techniques.

1. **Causal MIL** [21]: This algorithm utilizes the bag-level features as external variables to identify independent latent variables directly related to the prediction of each instance's feature. It then

Table 10: Performance differences for bags vs. instances in the MIL model.

| | | $\tau$ | 0.5 | 1 | 2 |
|---|---|---|---|---|---|
| Attention-Pooling | **Bag** | Macro-F1 | 0.8115 | 0.8486 | 0.8703 |
| | | Micro-F1 | 0.8152 | 0.8497 | 0.8710 |
| | | Weighted-F1 | 0.8163 | 0.8497 | 0.8710 |
| | | AUROC | 0.9567 | 0.9427 | 0.9555 |
| | Instance | Macro-F1 | 0.2912 | 0.3316 | 0.3418 |
| | | Micro-F1 | 0.2982 | 0.3407 | 0.3519 |
| | | Weighted-F1 | 0.2928 | 0.3368 | 0.3517 |
| | | AUROC | 0.7173 | 0.7224 | 0.7555 |
| | Performance differences | Macro-F1 | 0.5202 | 0.5170 | 0.5285 |
| | | Micro-F1 | 0.5170 | 0.5090 | 0.5191 |
| | | Weighted-F1 | 0.5235 | 0.5128 | 0.5193 |
| | | AUROC | 0.2394 | 0.2203 | 0.2000 |
| Additive-Pooling | Bag | Macro-F1 | 0.4867 | 0.4523 | 0.4938 |
| | | Micro-F1 | 0.5894 | 0.5560 | 0.5907 |
| | | Weighted-F1 | 0.5364 | 0.4904 | 0.5367 |
| | | AUROC | 0.9232 | 0.9133 | 0.9177 |
| | Instance | Macro-F1 | 0.2265 | 0.2387 | 0.2307 |
| | | Micro-F1 | 0.2709 | 0.1636 | 0.2777 |
| | | Weighted-F1 | 0.2575 | 0.0775 | 0.2713 |
| | | AUROC | 0.7666 | 0.8076 | 0.8533 |
| | Performance differences | Macro-F1 | 0.2602 | 0.2136 | 0.2631 |
| | | Micro-F1 | 0.3185 | 0.3924 | 0.3129 |
| | | Weighted-F1 | 0.2789 | 0.4130 | 0.2654 |
| | | AUROC | 0.1567 | 0.1058 | 0.0644 |
| Conjunctive-Pooling | Bag | Macro-F1 | 0.7956 | 0.7855 | 0.7937 |
| | | Micro-F1 | 0.7973 | 0.7867 | 0.7962 |
| | | Weighted-F1 | 0.7999 | 0.7880 | 0.7995 |
| | | AUROC | 0.9470 | 0.9442 | 0.9476 |
| | **Instance** | Macro-F1 | 0.6393 | 0.6422 | 0.6476 |
| | | Micro-F1 | 0.7664 | 0.7644 | 0.7702 |
| | | Weighted-F1 | 0.7977 | 0.7954 | 0.8012 |
| | | AUROC | 0.9503 | 0.9509 | 0.9537 |
| | **Performance differences** | Macro-F1 | 0.1563 | 0.1433 | 0.1461 |
| | | Micro-F1 | 0.0309 | 0.0223 | 0.0260 |
| | | Weighted-F1 | 0.0022 | -0.0074 | -0.0017 |
| | | AUROC | -0.0033 | -0.0067 | -0.0061 |

Table 11: Predicted performance for Snippets (i.e., bags) and patches (i.e., instances) of MD-MIL.

| | None-Attention | Attention | **Cross-Attention** |
|---|---|---|---|
| Snippet (Bag) | 0.87 | 0.88 | **0.91** |
| Patch (Instance) | 0.85 | 0.85 | **0.91** |

performs instance-pooling based on these latent variables. Since Causal MIL derives predictions using individual instance features and performs instance-pooling based on these values, it can be learnable on instances under the space $D_{XY}^{Ind}$.

2. **Multi-Instance Residual Evidential Learning (MIREL)** [38]: This algorithm is composed of two primary modules: 1) a module that predicts the bag label using the average feature of the instances that constitute the bag and 2) a module that performs predictions on individual instances based on their features. MIREL calculates the residual between the bag prediction and the instance predictions, which is indirectly used to perform instance-pooling. The instance prediction module of MIREL

operates based on the instance features themselves without applying weighted aggregation through additional information. Thus, it can be learnable on bag predictions under the space $D_{XY}^{Ind}$:

(a) The loss function of MIREL can be expressed as the sum of two loss terms generated by the bag and instance prediction modules:

- $L_{bag} = \ell_{bag}(\hat{y}_{bag}, y_{bag})$, $L_{inst} = \sum_{i=1}^{N} \ell_{inst}(\hat{y}_i, y_i)$
- $L_{total} = L_{bag} + \lambda L_{inst}$, where $\lambda$ is a weighting factor.
- $\hat{y}_{bag}$ is the prediction made by the bag prediction module; $y_{bag}$ is the true label for the bag; $\hat{y}_i$ is the label predicted by the instance prediction module; $y_i$ is the label inferred for each instance based on $\hat{y}_{bag}$ and residual evidence.

(b) The risks for the bag, instance, and total can be defined as follows:

- $R_{bag} = E_{(X,Y) \sim D_{XY}^{Ind}}[\ell_{bag}(\hat{y}_{bag}, y_{bag})]$
- $R_{inst_i} = E_{(X,Y) \sim D_{X_{inst_i} Y}^{Ind}}[\ell_{inst}(\hat{y}_i, y_i)]$
- $R_{total} = R_{bag} + \lambda \sum_{i=1}^{N} R_{inst_i}$

(c) Since our theoretical framework evaluates instance learnability under Assumption 1, we can express this as follows:

- $\inf R_{total} = \inf R_{bag} + \lambda \inf \sum_{i=1}^{N} R_{inst_i}$
- $R_{bag}$ is learnable under $D_{XY}^{Ind}$. Therefore, $\inf_{h \in H} R_{D_{XY}^{Ind}} = \inf R_{bag}$ holds.
- $\inf \sum_{i=1}^{N} R_{inst_i}$ is learnable with respect to the instance labels generated by the bag prediction module. Therefore, $\inf_{h \in H} R_{D_{XY}^{Ind}} = \lambda \inf \sum_{i=1}^{N} R_{inst_i}$.
- $\lambda$ is a constant, $\inf_{h \in H} R_{D_{XY}^{Ind}} = \inf \sum_{i=1}^{N} R_{inst_i}$. Therefore, Condition 4 holds.

Therefore, the algorithm is learnable for instances.

3. **Loss-based Attention for Deep Multiple Instance Learning (Loss-Attention)** [39]: This algorithm directly connects the attention mechanism with the loss function to simultaneously learn instance weights, instance predictions, and bag predictions. Specifically, Instance weights are calculated based on the loss function. The parameters of the fully connected layer for bag prediction are shared with the instance weight calculation. A regularization term consisting of instance weights and cross-entropy functions is introduced to improve instance recall. A consistent cost is added to smooth the learning process. Through this approach, Loss-Attention can more effectively learn the importance of instances and improve bag classification performance compared to existing attention-based MIL methods.

Loss-Attention [39] aims to simultaneously optimize the hypothesis spaces for both instances and bags by sharing parameters between them during the learning process. However, in $D_{XY}^{Gen}$, according to Condition 8, while the Loss-Attention algorithm is learnable for bags, it is not learnable for instances.

4. **Self-Attention Attention-based MIL Pooling (SA-AbMILP)** [40]: This algorithm enhances the reflection of relationships between instances compared to traditional Attention MIL [2] by employing a self-attention mechanism to compute instance features, which are then used for attention-pooling. Since SA-AbMILP uses transformed instance features that reflect relationships with other instances via self-attention for prediction and attention-pooling, it does not satisfy Condition 10 and is not learnable for instances.

5. **Transformer-based MIL (TransMIL)** [19]: This algorithm employs a Transformer module to better capture the relationships between instances. The Transformer module extracts a cls token feature based on the information of instances that reflect their relationships with other instances. TransMIL performs prediction on the bag using the cls token feature, a form of attention-pooling. Since TransMIL uses positional encoding features, its learnability for instances cannot be guaranteed by Theorem 11. Moreover, its attention-pooling does not satisfy Condition 10 and is not learnable for instances.

## G Limitations and Future works

According to the MIL survey by Carbonneau et al. [57], recent major issues in MIL have been defined as label ambiguity [58], label noise [59, 60], complex bag composition [61, 62, 63], and handling non-i.i.d. data [64, 65]. This study has only addressed the learnability of instances and did

not cover the applicability of these issues. In future work, we plan to utilize the theoretical framework developed in this study to devise countermeasures for issues such as label ambiguity and label noise, aiming to enhance the performance of MIL algorithms through theoretical research. Furthermore, we will seek solutions for problems like complex bag composition and handling non-i.i.d. data based on the theoretical framework of MIL algorithms. Additionally, noting that existing MD-MIL algorithms [6, 20] have extended non-learnable MIL algorithms to multi-dimensional cases, we will devise new MD-MIL methodologies based on the learnable MIL algorithms by our proposed theoretical framework.

