# OpenReview forum: "Are Multiple Instance Learning Algorithms Learnable for Instances?"
_NeurIPS.cc/2024/Conference — NeurIPS 2024 poster_

### Official Review · Reviewer_K1fQ · 2024-06-21

**Soundness:** 2
**Presentation:** 1
**Contribution:** 2
**Rating:** 5
**Confidence:** 4

**Summary:**

This paper aims to discuss the theoretical learnability of multi-instance learning. It first gives the necessary conditions for bag-level learnability and instance-level learnability. Then it discusses how the results can be used to verify the learnability of existing MIL algorithms. Finally, some empirical experiments are adopted to validate the theoretical findings.

**Strengths:**

This paper discusses the theoretical aspects of multi-instance learning (MIL). As MIL has attracted increased interest for applications in medical image analysis and time series analysis, it is nice to have some time to pause the development of heuristic algorithms and think about the theoretical foundations.

**Weaknesses:**

1. Many statements are not rigorous and this significantly reduces readability. For example, in Condition 4 the text refers to that the optimal hypothesis for independent instance domain must equal the sum of hypotheses for the individual instances. However, the formula is about the risks and they are not equivalent. In theorem 4, it just says "the MIL algorithm" without specifying which algorithm or the defining characteristics of the algorithm, which is not rigorous.

2. For the general bag domain discussed in the paper,  it is not the multi-instance bags that allow dependency among instances as commonly discussed and adopted by previous literature. According to the paper, this is essentially a weighted sum of the instances. It is better to claim this as such, instead of claiming it as a "general" bag domain, which it is not.

3. The categorization of pooling methods, as discussed in the appendix, is not consistent with how they are used in the referenced paper. See questions for details.

4. The empirical validation is somewhat lacking. Although this is mainly a theoretical discussion, I don't think the empirical are well designed to support the theoretical claims.

**Questions:**

1. In Section A.2, the attention pooling is different from Ilse et al., 2018, which is the attention-based MIL that most follow-up works are based on. Instead, additive pooling is actually the attention mechanism in Ilse et al., 2018. Why?

2. I am not very clear about the multi-dimension MIL discussed briefly in 4.3.2. Can you explain this sentence "For instance, in video data, each frame is composed of patches in multi-dimensional structures for each frame dimension."?

3. In Table 2, the title is validation of theorem 4, but in the text it says validating theorem 5?

4. Also in Table 2, do you have results for more recently proposed instance pooling approach, such as [21]? This also applies to the results in Table 3.

**Limitations:**

Some limitations are discussed in the appendix. However, it seems to contain some grammatical mistakes which hinder understanding. For example, what is the meaning of this sentence: "to address the problem that current MD-MIL algorithms [6, 20] extend non-learnable algorithms for instances to multidimensional cases"?

---

> ### Author Rebuttal · Authors · 2024-08-07
>
> We will address their feedback in the following:
>
> **(W1)**
> - **Condition 4**: The interpretation provided in Condition 4 aligns with PAC Learning theory. Specifically, $\inf R$ denotes the infimum of the risk within the given hypothesis space, representing the optimal hypothesis's risk. Thus, Eq. (9) refers to the risk of the optimal hypothesis concerning $D^{Ind}_{XY}$, consistent with PAC learning theory as seen in previous works [1, 2].
> - **Theorem 4**: All theorems in this paper, including Theorem 4, are proven under Assumption 1, which states that any MIL algorithm must be learnable at the instance level. Consequently, Theorem 4 applies to all MIL algorithms satisfying Assumption 1.
>
> **(W2)**
> - Many previous studies assume all instances are independent. However, recent studies such as [3-6] consider relationships between instances, improving prediction performance. These relationships are common in Deep MIL, particularly in practical environments like images, natural language, and time series. Therefore, our study treats the Independent bag domain space $D_{XY}^{Ind}$ as a special case and uses the term ‘General’ to define a space including $D_{XY}^{Ind}$.
> - To reduce misunderstandings, we propose defining the Dependent Bag Domain Space $D_{XY}^{Dep}$, which includes relationships between instances, and defining the General Bag Domain Space as $D_{XY}^{Gen} = D_{XY}^{Ind} \cup D_{XY}^{Dep}$.
>
> **(W4)**
> - To enhance empirical support for our theoretical framework, we added comparisons between theoretical applications and empirical results for various Deep MIL algorithms. The additional experimental results are presented in Tables A and B of the Global Rebuttal. Please refer to item 1 in the Global Rebuttal for details.
> - Additionally, we conducted experiments assuming an MD-MIL (Multi-Dimensional Multiple Instance Learning) problem. The results, shown in Table C and item 2 of the Global Rebuttal, demonstrate that using information from instances in other bags during the computation of attention weights yields effective learning outcomes. This suggests our framework can guide future MIL research.
>
> **(Q1)**
> - According to Section 2.4 of Ilse et al. (2018) [7], attention pooling involves computing attention weights for each instance, multiplying these by instance features, summing the weighted features to obtain a bag-level feature, and then using this for bag classification. In contrast, Javed et al. (2022) [8] describe additive pooling, where predictions are made based on each instance's weighted features, and these individual predictions are summed for a bag-level prediction.
> - Our paper represents attention pooling as averaging the weighted features (Line 651), while additive pooling is represented by summing individual predictions (Line 655). Although our attention pooling formula includes a factor of $\frac{1}{N}$, this does not fundamentally change the operation. To prevent misunderstanding, we will remove this factor to align it exactly with [7].
>
> **(Q2)**
>
> - The sentence provides an example to understand Multi-Dimensional MIL (MD-MIL). MD-MIL extends the traditional MIL problem to make predictions for bags-of-bags, using the outer bag's label to infer inner bags and instances.
> - For example, in video anomaly detection (VAD), the goal is to detect the exact frame with an anomaly based on the video label. By interpreting VAD as an MD-MIL problem, each frame can be considered a bag of image patches, and the video as a bag-of-bags.
> - We conducted experiments on such cases, as detailed in item 3 of the Global Rebuttal. Table C shows that MD-MIL algorithms satisfying our framework's theoretical conditions achieved the best performance.
>
> **(Q3)**
>
> - You are correct, the title should be "Validation of Theorem 5." We have reviewed all similar instances and will correct this error in the final paper version.
>
> **(Q4)**
>
> - The results for the requested instance-pooling algorithm [21] are included in Table A of the Global Rebuttal PDF. These results show that other recent instance-pooling algorithms are also not learnable on $D_{XY}^{Gen}$ as they are on $D_{XY}^{Ind}$.
> - Applying these results to Table 3, we found that if an algorithm is not learnable for bags, it is also not learnable for instances, as confirmed by our experimental results in line with Theorem 1.
>
> **(Limitation 1)**
>
> - The sentence explains that current MD-MIL research adopts pooling methods that are not learnable for instances according to our theoretical framework. Specifically, embedding-pooling and attention-pooling methods used in [9] and [10] are not learnable for instances, making the MD-MIL algorithms using these methods also unlearnable.
> - We will revise and clarify these points to ensure they accurately reflect the limitations and are grammatically correct.
>
> **References**
>
> [1] Mohri et al., *Foundations of Machine Learning*, MIT Press, 2018.
>
> [2] Fang et al., "Is out-of-distribution detection learnable?", *NeurIPS*, 2022.
>
> [3] Angelidis et al., "Multiple instance learning networks for fine-grained sentiment analysis", *TACL*, 2018.
>
> [4] Shao et al., "TransMIL: Transformer based correlated multiple instance learning for whole slide image classification", *NeurIPS*, 2021.
>
> [5] Early et al., "Inherently interpretable time series classification via multiple instance learning", *ICLR*, 2024.
>
> [6] Chen et al., "TimeMIL: Advancing Multivariate Time Series Classification via a Time-aware Multiple Instance Learning", *ICML*, 2024.
>
> [7] Ilse et al., "Attention-based deep multiple instance learning", *PMLR*, 2018.
>
> [8] Javed et al., "Additive MIL: Intrinsically interpretable multiple instance learning for pathology", *NeurIPS*, 2022.
>
> [9] Tibo et al., "Learning and interpreting multi-multi-instance learning networks", *Journal of Machine Learning Research*, 2020.
>
> [10] Fuster et al., "Nested multiple instance learning with attention mechanisms", *ICMLA*, 2022.

---

> > ### Comment · Reviewer_K1fQ · 2024-08-09
> >
> > Thanks for the response. Some of my questions/concerns have been addressed, but I think you misunderstood some of my original comments.
> >
> > For example, regarding W1, a hypothesis is a function while a hypothesis space is a set of functions. In the paper, it was written as "the optimal hypothesis"... "equal to the sum of hypotheses", which refers to the summation of functions. But Eq 9 refer to the summation of risks (of the functions). They are not the same.

---

> > > ### Author Response · Authors · 2024-08-09
> > >
> > > Thank you for your feedback. As you correctly pointed out, Condition 4 contains inaccuracies in its expression. While the formula in Condition 4 is accurate, the phrase "the optimal hypotheses" should be revised to "the risks of the optimal hypotheses."
> > >
> > > According to Definition 6, we proposed Conditions 4 and 7 as necessary prerequisites for satisfying Condition 2. We have demonstrated, through mathematical proofs based on these conditions, that if Conditions 4 and 7 are met, then Condition 2 is also satisfied. We have confirmed that the formulas and proofs are presented correctly. In light of your comments, we plan to revise expressions in both Condition 4 and Condition 7 as follows (Modifications are represented in bold.):
> > >
> > > Condition 4: **The risk of** the optimal hypothesis for $D_{XY}^{Ind}$ must ensure that it equals the sum of **the individual risks** of the optimal hypotheses within $D_{XY}^{Ind}$:
> > >
> > > Condition 7: **The risk of** the optimal hypothesis for $D_{XY}^{Gen}$ must ensure that it equals the weighted sum of **the individual risks** of the optimal hypotheses within $D_{XY}^{Gen}$:
> > >
> > >
> > > We will make these revisions to address the errors and improve the clarity of our paper. If you have any further questions or concerns, please let us know to clarifity them as well. Thank you again for your valuable feedback.

---

> > > > ### Comment · Reviewer_K1fQ · 2024-08-11
> > > >
> > > > Thanks for the follow-up response and your commitment to revising the paper. Scores has been revised.

---

> > > > > ### Author Response · Authors · 2024-08-12
> > > > >
> > > > > Thank you again for your valuable feedback. We will do our best to reflect your feedback on the final version of the paper. If you have any further questions or concerns, please let us know so that we can respond to them accordingly.

---

### Official Review · Reviewer_bVsY · 2024-07-03

**Soundness:** 3
**Presentation:** 3
**Contribution:** 3
**Rating:** 6
**Confidence:** 5

**Summary:**

This paper mainly studies the instance-level learnability of common weakly-supervised MIL algorithms. With the PAC theoretical framework, it shows the conditions for MIL algorithms to be learnable for instances. Two general cases of instance distribution, IID instances and non-IID instances, are discussed and analyzed in the proposed theoretical framework. In addition, the proposed theoretical framework covers a wide range of MIL algorithms, which could provide valuable insights for many MIL-based applications in the real-world.

**Strengths:**

- This paper is overall well-written and easy to follow. All key definitions and theorems’ implications are given and explained clearly.
- This paper studies an interesting and valuable problem, *i.e.*, instance-level learnability in weakly-supervised MIL. This problem is still in the stage of empirical exploration, so a formal study from a theoretical perspective, which this work presents, is needed and could be valuable to the community of deep MIL.
- The proposed theoretical framework covers a wide range of MIL algorithms, which could provide valuable insights for many MIL-based applications in the real-world.

**Weaknesses:**

- In Definition 6, the sufficient condition for instance-level PCA learnability, i.e., Condition 2, is given. However, the authors do not prove that Condition 2 is necessary. From my understanding, the instance-level PCA learnability cannot directly deduce Condition 2, as it (Eq. 7) contains an additional bag-level constraint.
- The authors mainly study one type of MIL algorithm in which pooling (instance embedding level or instance score level) is only performed once. In fact, there is another type of MIL algorithm with multiple pooling operations [1, 2, 3]. Most of these algorithms aim at making instance-level predictions and could be viewed as a combination of the classical algorithms discussed in the paper. The authors are encouraged to discuss them and indicate whether these MIL algorithms could also be incorporated into the proposed theoretical framework.

Minor issues:
- Table 3: Please consider using a clearer way to present the performance of instance-level prediction and bag-level prediction, as well as their difference.
- To meet the basic publication requirements of NeurlPS, the authors are encouraged to carefully and repeatedly check those easy-to-avoid errors in their paper, *e.g.*, line 346 - “pr,edication” and “Additve” on page 33.

References:

[1] Shi et al., Loss-based attention for deep multiple instance learning. AAAI, 2020.

[2] Liu et al., Weakly-Supervised Residual Evidential Learning for Multi-Instance Uncertainty Estimation. ICML, 2024.

[3] Wang et al., Rethinking Multiple Instance Learning for Whole Slide Image Classification: A Bag-Level Classifier is a Good Instance-Level Teacher. IEEE Transaction on Medical Imaging, 2024.

**Questions:**

See my comments above.

**Limitations:**

Limitations have been indicated by the authors and written in the paper. There is no additional limitation that needs to be explicitly highlighted here.

---

> ### Author Rebuttal · Authors · 2024-08-07
>
> We appreciate your recognition and feedback on our work. We've integrated your suggestions to enhance our manuscript. For more information, see Global Response. We are addressing your specific comments as follows:
>
> **(W1)** Reason for Condition 2 Being a Necessary Condition for Instance-Level Learnability
>
> - According to Theorem 1, if a model is not learnable at the bag level, it cannot be learnable at the instance level. Therefore, for a model to be learnable at the instance level, learnability at the bag level must be ensured, making Condition 2 a necessary condition for instance-level learnability. In response to the reviewer's comment, we will add a clear explanation in Definition 6 to improve clarity.
>
> **(W2)** Applicability of the Framework to Various Latest Deep MIL Algorithms
>
> - Our study aims to propose a theoretical framework that serves as a baseline for MIL. Therefore, we did not discuss all methods that use various strategies for pooling to enhance performance. However, Theorems 1, 2, and 3, along with Theorems 4-12, which verify the learnability of different pooling methods at the instance level, can be applied to most MIL algorithms. Specifically, for the special MIL algorithms mentioned by the reviewer [1, 2, 3], we can determine instance-level learnability as follows:
>     - **[1]**: The MIL algorithm proposed in [1] uses an attention mechanism to derive features for the bag from the features of each instance. It then makes predictions at the bag level and uses attention to make instance-level predictions. Thus, because the algorithm in [1] does not satisfy Condition 10 of Theorem 8, it is not learnable at the instance level.
>     - **[2, 3]**: The methodologies in [2, 3] obtain predictions for instances based on their individual features and then use these features to make bag-level predictions through an independent classifier. The predictions for instances are trained to approximate the bag-level predictions, making these algorithms a type of instance-pooling. Since the optimal hypothesis space for individual instances approximates the optimal hypothesis space for bags, these MIL algorithms are learnable at the instance level according to Theorem 6.
> - As mentioned in item 1 of the Global Rebuttal, we have added detailed analyses and indications for various latest Deep MIL algorithms and will include these in the paper. Furthermore, we will strengthen the robustness of our theory through empirical validation of recent MIL algorithms, as shown in Tables A and B of the attached PDF.
>
> **(Minor Issues)**
>
> - To make the experimental results in Table 3 clearer, we will revise the table to separately show the Macro-F1 Score and AUROC results, and display the bag-level and instance-level prediction results together, as shown in Table B of the attached PDF. This should provide clearer performance metrics.
> - We have reviewed and corrected the mentioned typographical errors and will reflect these corrections in the paper.
>
> **References:**
>
> [1] Shi et al., Loss-based attention for deep multiple instance learning. AAAI, 2020.
>
> [2] Liu et al., Weakly-Supervised Residual Evidential Learning for Multi-Instance Uncertainty Estimation. ICML, 2024.
>
> [3] Wang et al., Rethinking Multiple Instance Learning for Whole Slide Image Classification: A Bag-Level Classifier is a Good Instance-Level Teacher. IEEE Transaction on Medical Imaging, 2024.

---

> > ### Comment · Reviewer_bVsY · 2024-08-09
> > **Reply to the Author's rebuttal**
> >
> > Thanks for the author's efforts and detailed responses. After reading the rebuttal, I still have some concerns as follows:
> > - **W1**: The authors mention that Theorem 1 makes Condition 1 a necessary condition for instance-level PAC learnability since instance-level learnability implies bag-level learnability according to Theorem 1. So, does this mean that Condition 2 can be simplified, *i.e.*, just removing the bag-level constraint from Eq. (7)? Further, if Condition 2 can be simplified as I said, what is the difference between Eq. (7) and Eq. (4)? From my understanding, they are the same. That is to say, Condition 2 seems meaningless and not mentionable as it is just a straightforward variant of Definition 5.
> > - **W2**: For [2,3], I don't think the optimal hypothesis space for individual instances approximates the optimal hypothesis space for bags. Could the authors explain or justify this?
> > - **Experimental results in Table A**: I wonder how the authors implement the bag-level prediction for the methods [1,2]. To my knowledge, all of [1,2,3] make bag-level predictions using their respective bag-level branch.

---

> ### Author Response · Authors · 2024-08-10
> **(W1)**
>
> **Answer:**
>
> Thank you for taking the time to thoughtfully read our study and rebuttals. Please find the additional answers to your further questions and concerns.
>
> **(W1)**
>
> - Condition 2 is a necessary condition to verify that for a MIL algorithm to be learnable at the instance level.
>     - Under Assumption 1, if it is learnable at the bag level, Condition 2 ensures learnability at the instance level.
>     - Hence, this is an essential condition defining the relationship between the hypothesis spaces of the bag and the instance, which is the theoretical foundation for proving the learnability of instances in our study.
> - The differences between Eq. (4) and Eq. (7) are as follows:
>     - Eq. (4) deals with the definition of learnability for individual instances by a MIL algorithm.
>         - Since MIL is a problem of learning instances based on bag-level labels, learnability at the instance level alone does not sufficiently explain the algorithm's ability to learn instances.
>     - On the other hand, Eq. (7) defines the necessary and sufficient condition for a MIL algorithm to be learnable at the instance level, which is that the relationship between instance-level and bag-level learnability must be "equivalent."
>         - Therefore, the proofs of Conditions 4 and 7 in Section 3, which are necessary for each MIL algorithm to be learnable at the instance level, are based on Eq. (7) in Condition 2.
>
> To clarify the necessity and significance of Condition 2, distinct from Eq. (4), we will reflect this in the paragraph after Condition 2 to improve the paper.

---

> ### Author Response · Authors · 2024-08-10
> **(W2)**
>
> **(W2)**
> - The reasons why the methods proposed by Liu et al. [2] and Wang et al. [3] can be interpreted through the theoretical framework proposed in this study are explained as follows:
> - **Liu et al. [2]**
>     - MIREL (Multiple Instance Regression with Embedding Learning) [2] is composed of two main modules:
>         1. A module that extracts features for the bag by mean-pooling the features of instances within the bag and then predicts the label for the bag based on these features.
>         2. A module that predicts the label for each instance using the individual features of the instances within the bag.
>     - Additionally, MIREL includes a residual evidence module that calculates the difference between bag and instance predictions, ensuring that the instance and bag prediction modules work complementarily. This means the instance prediction module learns based on the bag prediction value.
>     - The instance prediction module performs based on the instance’s own features and does not use weighted sums from additional information. Therefore, MIREL's instance prediction module can learn for the bag on $D_{XY}^{Ind}$.
>     - The learnability of MIREL’s instance prediction module, satisfying Condition 4 of our framework, can be explained as follows:
>         1. The loss function of MIREL can be expressed as the sum of the loss functions generated by the two main prediction modules for the bag and instance:
>             - $L_{bag}$$=\ell_{bag}$ $\(\hat{y_{bag}}, y_{bag}\) $ , $L_{inst} = \sum_{i=1}^{N} \ell_{inst}(\hat{y_i}, y_i)$, where $\hat{y_{bag}}$ is the label predicted by the bag prediction module, $y_{bag}$ is the actual label of the bag, $\hat{y_i}$ is the label predicted by the instance prediction module, and $y_i$ is the label of the instance obtained from $\hat{y_{bag}}$ and residual evidence
>             - $L_{total} = L_{bag} + \lambda L_{inst}$, where $\lambda$ is a weight
>         2. The risk for each bag, each instance, and total can be defined as follows:
>             - $R_{bag} = E_{(X, Y) \sim D_{XY}^{Ind}}[\ell_{bag}(\hat{y}_{bag}, y_{bag})]$
>             - $R_{inst_i} = E_{(X, Y) \sim D_{X_{inst_i}Y}^{Ind}}[\ell_{inst}(\hat{y}_i, y_i)]$
>             - $R_{total} = R_{bag} + \lambda \sum_{i=1}^{N} R_{inst_i}$
>         3. Since our theoretical framework evaluates learnability at the instance level under Assumption 1, it can be expressed as follows:
>             - $\inf R_{total} = \inf R_{bag} + \lambda \inf \sum_{i=1}^{N} R_{inst_i}$
>             - Since $R_{bag}$ is learnable on $D_{XY}^{Ind}$, $\inf_{h \in H} R_{D_{XY}^{Ind}} = \inf R_{bag}$ holds.
>             - $\inf \sum_{i=1}^{N} R_{inst_i}$ learns from the instance labels generated by the bag prediction module, so $\inf_{h \in H} R_{D_{XY}^{Ind}} = \lambda \inf \sum_{i=1}^{N} R_{inst_i}$.
>             - Since $\lambda$ is a fixed constant, $\inf_{h \in H} R_{D_{XY}^{Ind}} = \inf \sum_{i=1}^{N} R_{inst_i}$ (i.e., Condition 4) holds.
> - **Wang et al.[3]**
>     - Unlike instance pooling [2], we have confirmed that ICMIL (Iteratively Coupled Multiple Instance Learning) [3] employs an attention-pooling method. Therefore, we need to explain how our theoretical framework interprets ICMIL separate from [2].
>     - ICMIL is an algorithm where a teacher MIL model performs attention-pooling to pseudo label each instance based on its attention confidence score, followed by a student model performing supervised learning using these pseudo labels.
>     - The teacher MIL model assigns an attention weight between 0 and 1 to each instance, with the sum of these weights equal to 1. This space, multiplied by the hypothesis space of each instance, satisfies Condition 8 and, according to Theorem 5, is learnable for the bag on $D_{XY}^{Gen}$. This can be verified through proofs in Appendix C.5.
>     - Therefore, the hypothesis space for instances in the teacher model of ICMIL is included in the hypothesis space for the bag-level generated space of individual instances. However, since it does not comply with Condition 10, as proven in Appendix C.9, we cannot guarantee that the teacher model in ICMIL is learnable for instances.
> - We have rigorously presented terms, definitions, and theories related to Deep MIL algorithms based on PAC theory to cover all the MIL algorithms under Assumption 1. We will improve the paper by including the cases of [2] and [3] as representative examples how our framework interprets a specific MIL algorithm.

---

> ### Author Response · Authors · 2024-08-10
> **(Experimental results in Table A)**
>
> **(Experimental results in Table A)**
>
> - **Shi et al.[1]**
>     - [1] addresses the issue where the attention mechanism used in attention-pooling assigns high weights to irrelevant instances, leading to incorrect predictions through a loss-based attention mechanism. [1] performs predictions for both instances and bags simultaneously, and we can measure its performance in the same way with other MIL models that use attention-pooling.
> - **Liu et al. [2]**
>     - MIREL [2] separates the prediction modules for the bag and the instances. In our study, in accordance with Condition 2, we aim to evaluate the prediction performance for the bag based on the prediction results for the instances. Therefore, instead of directly utilizing the performance of the prediction module for the bag, we calculated the bag predictions by applying mean-pooling and max-pooling on the scores from the instance prediction module, and then measured the average performance.
>     - This experimental design was intended to demonstrate that MIREL[2] shares characteristics with other instance-pooling methods in the case for the bag-level prediction based on the results of the instance-level predictions.
>     - Now, as shown in (W2), we verified that MIREL[2] can indeed be interpreted as an instance-pooling method, and we have also sufficiently experimented with other instance-pooling methods in Table A. To avoid any unnecessary misunderstandings as pointed out, we will exclude the results from Table A.
>
> Thanks again for your rigorous feedback, which helps us to improve the quality of the paper. If you have further concerns and questions, please let us know so that we can respond to them.

---

> ### Comment · Reviewer_bVsY · 2024-08-12
> **Reply to the Authors' Rebuttal**
>
> Thanks for the authors' comprehensive responses. These have addressed my concerns raised before. The authors are encouraged to include their responses (to Condition 2 and the theoretical interpretation and experimental settings of [1,2]) into the final version of the paper.
> ﻿
> In view of these, I am glad to keep my positive score. I believe this paper would impact the theoretical study of instance-level learnability in MIL.

---

> > ### Author Response · Authors · 2024-08-12
> >
> > We deeply appreciate your positive evaluation of our paper and your kind acknowledgment of our comprehensive responses. We are pleased to have addressed your all concerns. We will incorporate your suggestions into the final version of the paper, including our responses to Condition 2 and the theoretical interpretation and experimental settings of [1,2]. We believe that this will enhance the contribution and quality of the paper. Thank you once again for your valuable feedback, and please do not hesitate to let us know if you have any additional questions or concerns.

---

### Official Review · Reviewer_W4rs · 2024-07-08

**Soundness:** 4
**Presentation:** 1
**Contribution:** 4
**Rating:** 5
**Confidence:** 4

**Summary:**

The paper provides theoretical considerations regarding Multiple Instance Learning.

**Strengths:**

Theoretical consideration supported by the experimental results.

**Weaknesses:**

There are no images depicting the intuition behind those definitions and theorems.

Paper is extremely hard to follow because it is not giving the intuition and implication for practitioners.

**Questions:**

Is it possible to improve presentation of the paper?

**Limitations:**

Discussed.

---

> ### Author Rebuttal · Authors · 2024-08-07
>
> We appreciate your recognition and feedback on our work. We've integrated your suggestions to enhance our manuscript. For more information, see the Global Response. We are addressing your specific comments as follows:
>
> **(W1, 2, Q1)** Improving the Presentation of the Paper
>
> - To address the feedback that the paper is difficult to understand due to its focus on theoretical content and lack of figures, we will add example figures illustrating the problems addressed in our study. Additionally, we will include Figure A to summarize the relationships between the theorems.
>     - **Attached PDF Figure A:** Figure A summarizes the relationships between all the theorems that constitute the theoretical framework of our paper. It not only shows which theorems are derived from others but also helps to identify which MIL pooling methods or algorithms are learnable or not at the instance level. Figure A will help readers understand the theoretical framework by showing the connections and outcomes between the theorems.
> - Furthermore, since the paper focuses primarily on theoretical validation, we have significantly strengthened the empirical validation to show how the framework can be applied in practical environments, as detailed in items 1 and 2 of the Global Rebuttal. This will help readers understand the real-world impact of our framework.
> - Finally, we will enhance the presentation by thoroughly reviewing and correcting grammatical errors, table captions, typos, and other issues identified by reviewers.

---

> > ### Comment · Reviewer_W4rs · 2024-08-09
> >
> > Thank you for your response; it addressed my concerns. Additionally, I recommend discussing this work in the context of the theory you have developed, as I found it intriguing. It introduces a novel MIL assumption, which adds significant value.
> >
> > Struski, Łukasz, et al. "ProMIL: Probabilistic multiple instance learning for medical imaging." ECAI 2023. IOS Press, 2023. 2210-2217.

---

> > > ### Author Response · Authors · 2024-08-10
> > >
> > > Thanks for adjusting your score. To further respond to your valuable feedback, we would like to add the discussion for ProMIL in the paper according to the following analysis.
> > >
> > > ProMIL learns from individual instance predictions by performing predictions on each instance and then uses the classification results of instances within a specified quantile to make predictions for the bag using a Bernstein Polynomial Estimator. This approach enables more accurate quantile estimation in datasets with complex or uneven distributions, thereby improving the predictive performance for the bag.
> > >
> > > - Let the prediction module for an instance be $f(x_i)$, and let the module's prediction value be $c_i$ (where $0 \leq c_i \leq 1$). The algorithm's prediction value for the bag, $c_q$, based on the $c_i$ values within the *q*-quantile, is given by:
> > >     - $c_q = \sum_{k=0}^{n} \binom{n}{k} q^{n-k} (1-q)^k \cdot c_k$
> > > - If $c_q$ is greater than 0.5, the bag is predicted to be positive; otherwise, it is predicted to be negative.
> > >
> > > ProMIL can be interpreted through our proposed theoretical framework as follows:
> > >
> > > - ProMIL makes predictions for the bag based on individual instance predictions. Thus, it does not satisfy Condition 10 of our framework and can only be learned on $D_{XY}^{Ind}$, not on $D_{XY}^{Gen}$.
> > > - According to our Definitions 2 and 4, the relationship between ProMIL and the model's risk on $D_{XY}^{Ind}$ can be expressed as:
> > >     - $\inf R_{bag} = \inf \left( \sum_{k=0}^{n} \binom{n}{k} q^{n-k} (1-q)^k \cdot R_{inst_k} \right)$
> > > - According to our Assumption 1, if learning on the bag is possible, the following holds:
> > >     - $\inf_{h \in H} R_{D_{Ind}^{XY}} = \inf \left( \sum_{k=0}^{n} \binom{n}{k} q^{n-k} (1-q)^k \cdot R_{inst_k} \right) = \sum_{i=1}^{N} \inf R_{inst_i}$
> > > - Therefore, ProMIL ensures satisfaction of Condition 4 by optimizing the total bag risk through the sum of each instance's optimal risk. This demonstrates that learning on instances is feasible on $D_{XY}^{Ind}$.
> > >
> > > These analytical results illustrate that our framework based on PAC Learning theory can effectively analyze specific real-world algorithms like ProMIL. This serves as a representative example demonstrating the utility of our proposed framework.
> > >
> > > Thank you once again for your valuable feedback. If you have any further questions or concerns, please let me know so as to further respond to them.

---

### Official Review · Reviewer_dSpL · 2024-07-13

**Soundness:** 3
**Presentation:** 3
**Contribution:** 2
**Rating:** 6
**Confidence:** 4

**Summary:**

The main contributions of the paper are theoretical:
- Proof of the fact that MIL algorithms that are not learnable for bags do not guarantee learnability for instances.
- Using PAC learning theory (under some assumptions) learnability conditions are derived. These conditions are sufficient and necessary.
- Basic experiments are carried out to support the theoretical findings empirically.

The most important results of the paper are theorems 1, 3, 5, 7, 10, 11.

**Strengths:**

The strongest part of the paper is the theoretical framework and mathematical setup resulting in some key results about PAC learnability at the instance level for various MIL algorithms.
- Explicit results connecting bag learnability with instance learnability.
- The paper derives theoretical guarantees about which kinds of pooling maintain PAC learnability for instances.
- I think the paper could have broader impact in the development of instance level MIL methods.
- The paper despite being quite dense with theorems and definitions flows reasonably well and is understandable when read. I checked the proofs and I can follow the arguments line by line. I appreciate that the authors take the time to set up clear notation and an proper mathematical formulation of their problem definition.

**Weaknesses:**

Despite the strengths of the paper there are some weaknesses.
- From a motivation standpoint, it has been the case for a while now that the strongest methods in MIL are attention pooling based and operate at the bag/embedding level, not the individual instance level. How do the results presented in this paper affect methods such as [1]? Why should we care about instance based methods in 2024? Explainability or interpretability motivations? The paper should make the case for why these results are important given the current state-of-the-art in the field.
- The methodology is well supported by the proofs. However, the empirical evaluation is not very thorough. It would be nice to see if the methodology has practical implications in standard MIL benchmarks used in new methods. Datasets such as the ones used in [1] are basically standard for MIL research. If the authors want to go one step further they could incorporate more elaborate testing frameworks too [2]. More experiments would help to convince me that the theoretical results have grounding in empirical findings and encourage me to raise my score.

Refrences:

[1] Attention-based Deep Multiple Instance Learning, Maximilian Ilse, Jakub M. Tomczak, Max Welling, ICML (2017).
[2] Reproducibility in multiple instance learning: a case for algorithmic unit tests., Raff, Edward, and James Holt., NeruIPS (2024).

**Questions:**

- The assumptions (referred to as conditions) 4, 6, 7 are crucial for the theorem proofs in the paper. Are these conditions assumed to be true? Or are they proven to be true somewhere? Is there a way to check if these conditions are true for a given model / dataset practically?

- The framework analyzes independently distributed instances within a bag $\mathcal{D}^{\text{Ind}}_{XY}$  as well as cases where instances within the same bag are statistically dependent. However, in many cases it is important to model the relation between different bags (or instances inside different bags). For example, thinking of an image as a set of bags where each bag (patch of pixels) is a set of feature instances (pixels), we might want to learn relationships between nearby pixels that fall in different bags. Do the authors have any insight on if their analysis could be expanded in this direction in future work?

**Limitations:**

The paper discusses its limitations.

---

> ### Author Rebuttal · Authors · 2024-08-07
>
> We appreciate your recognition and feedback on our work. We've integrated your suggestions to enhance our manuscript. For more information, see the Global Rebuttal. We are addressing your specific comments as follows:
>
> **(W1)**
> - **Importance of Instance Pooling:**
>     - Instance-Pooling is a fundamental method still used in recent studies [2-4].
>         - The Independent bag domain space assumes instance independence, common in medical domains and video anomaly detection. In such cases, Instance-Pooling is effective due to lower computational and memory requirements compared to other pooling methods.
>         - For example, Causal MIL (Neurips, 2022) improves prediction by identifying influential features while assuming instance independence within a bag on $D_{XY}^{Ind}$.
> - **Learnability of Attention MIL[1]:**
>     - Our paper presents Theorem 8 to evaluate the learnability of instance-based attention pooling, a popular MIL method. According to Theorem 8, if the hypothesis space for bags exceeds that of instances, Attention MIL (Pooling) is not learnable at the instance level. This conclusion applies to methods like [1], TransMIL [6], SA-AbMIL [7], and TimeMIL [8].
>     - Our framework also evaluates other attention-based methods like Additive MIL and Conjunctive MIL. We will include Figure A in the Global Rebuttal PDF for better understanding.
>
> **(W2)**
> - Our study proposes a framework to overcome the limitation of previous studies focusing only on experiments without theoretical validation of instance learnability. We prioritized empirical verification of pooling methods theories rather than specific algorithm results.
>     - The benchmark from [1], mentioned by the reviewer, is based on real datasets where generalization is not assumed, making theoretical results difficult to observe. Various algorithms have already been tested on these datasets.
>     - The benchmark from [5] confirms reproducibility of bags composed of vector datasets sampled from a normal distribution on the Independent bag domain space, evaluating MIL algorithm adherence to MIL assumptions. Our study differs by proposing a framework for evaluating instance learnability and defining the General Bag Domain space to assess learnability in environments with instance relationships.
>     - The synthetic dataset using MNIST data, similar to [2], is widely used for specific cases. It accurately verifies the theory without performance variations due to encoder differences, which is why we used it.
> - Reflecting the reviewers' suggestion for more empirical evaluations, we conducted verifications on various latest Deep MIL algorithms in the Global Rebuttal. The results, shown in Tables A and B, align with our theoretical results and will be included in the revised paper.
>
> **(Q1)**
> - For Conditions 4 and 7, we have proven they are necessary and sufficient based on Definition 6 in our paper, as shown in Appendix C.2 and C.3. Conditions 4 and 7 hold true for any model as long as Assumption 1 is satisfied. Specifically, experimental validation of Theorems 8-11, derived from Theorem 3, shows that Condition 7 is necessary and sufficient for learnability at the instance level on $D_{XY}^{Gen}$.
> - Condition 6 is assumed to be true in our paper. When relationships exist between instances, their contributions to the bag's domain space vary. The total contribution sum is 1. For example, in the **Experimental Validation of Theorems 5, 8, and 9**, if the digits 3 and 5 appear together in a bag, they are positive, but individually they are not. Thus, contributions change based on relationships.
>
> **(Q2)**
> - The scenario you mentioned relates to Multi-Dimensional MIL (MD-MIL). For example, in video data, a video is composed of snippets, and each snippet consists of image patches. For video anomaly detection, information from a snippet’s patches may require information from previous snippets’ patches.
> - Our study clarified that to make predictions at the instance level using relationships between instances, Condition 10 must not be violated, as stated in Theorem 8. External information should not be used in the prediction stage itself but only in the calculation of weights, such as attention weights.
> - We conducted additional experiments on such cases, as detailed in item 2 of the Global Rebuttal. The results confirmed that theoretically learnable algorithms achieved the best performance, demonstrating the potential for extending our theoretical framework to various domains. These findings will be included in the revised paper.
>
> **References:**
>
> [1] Ilse et al., Attention-based deep multiple instance learning. PMLR., 2018.
>
> [2] Zhang et al., Multi-instance causal representation learning for instance label prediction and out-of-distribution generalization. Neurips, 2022.
>
> [3] Liu et al., Weakly-Supervised Residual Evidential Learning for Multi-Instance Uncertainty Estimation. ICML, 2024.
>
> [4] Wang et al., Rethinking Multiple Instance Learning for Whole Slide Image Classification: A Bag-Level Classifier is a Good Instance-Level Teacher. IEEE Transaction on Medical Imaging, 2024.
>
> [5] Raff et al., Reproducibility in multiple instance learning: a case for algorithmic unit tests., Neurips, 2024.
>
> [6] Shao et al., Transmil: Transformer based correlated multiple instance learning for whole slide image classification. Neurips, 2021.
>
> [7] Rymarczyk et al., Kernel self-attention in deep multiple instance learning. arXiv, 2020.
>
> [8] Chen et al., TimeMIL: Advancing Multivariate Time Series Classification via a Time-aware Multiple Instance Learning. ICML, 2024

---

> > ### Comment · Reviewer_dSpL · 2024-08-11
> >
> > I would like to thank the authors for the thorough response.
> >
> > Most of my concerns were addressed and thus I am raising my score.
> >
> > Though the paper is technically solid I still have some doubts about the impact. I understand that this is my subjective opinion of the work and encourage the authors to expand on how they think their work might impact subsequent research.

---

> ### Author Response · Authors · 2024-08-12
>
> Thanks for your positive considerations and valuable further concerns for the subsequent research. We believe the answers will be helpful in clarifying the implications of our work. Therefore, we eagerly address your concerns as follows..
>
> 1. The theoretical framework proposed in this study can verify whether a specific MIL algorithm is learnable on instances for all MIL algorithms only if they satisfy Assumption 1.
>     - We derived Condition 1, a necessary and sufficient condition for learning on instances for MIL that satisfies Assumption 1, and through proofs, we demonstrated that all MIL algorithms satisfying Condition 1 can be applied to the proposed framework.
>     - As concrete examples, in additional responses to other reviewers (**bVsY(W2)**, **W4rs**), we showed that specific MIL algorithms such as MIREL, ICMIL, and ProMIL can be interpreted using the proposed framework. By demonstrating how the proposed framework can be applied to actual MIL with these representative cases, we aim to improve the paper.
> 2. It enables the appropriate selection of MIL algorithms across various real-world applications where MIL can be used.
>     - MIL is utilized in fields such as Medical Image Analysis, Video Anomaly Detection, and Biological Activity Modeling, where detailed labeling is difficult but accurate prediction for instances (e.g., cancer-affected regions, anomaly detection time points, molecules constituting new drugs) is essential. These applications include cancer diagnosis, enhancing the efficiency of security and surveillance systems, and drug development.
>     - Particularly in the medical field, where accurate prediction for instances is required, Attention-pooling-based MIL has been utilized recently due to its high prediction performance on bags [1,2,3,4]. However, according to our theoretical validation of the framework, Attention-pooling does not satisfy Condition 10 and therefore is not learnable for instances. This has been experimentally verified through various Attention-pooling methods in Table B of Global Rebuttal 1. Consequently, these algorithms may yield inaccurate predictions for instances, indicating the need for the introduction of learnable pooling methods for instances, such as Conjunctive-Pooling.
> 3. Theorem 8-11 and Condition 9 can provide direction for future MIL research in more practical problem settings.
>     - In the second item of Global Rebuttal and Table C, we demonstrated that Multi-Dimensional-MIL (MD-MIL), which combines MIL algorithms satisfying Theorem 8-11 and Condition 9, achieves the best performance in tasks predicting multi-dimensional instances.
>         - MD-MIL assumes a more practical problem by considering not only the relationships between instances within a bag, which most current MIL algorithms consider, but also the relationships between bags. Although this is still in the early stages of research, it is expected to be actively studied in the future. In Section 4.3.2, we showed that MD-MIL can be interpreted through our framework, and its potential has been experimentally proven in the rebuttal.
>         - These results demonstrate the scalability of the proposed framework and provide a theoretical basis for future research in MD-MIL.
>     - Additionally, Theorem 8-11 and Condition 9 theoretically interpret how external information to instances should be applied in MIL algorithms and provide direction for practical application.
>         - For example, in tasks predicting cancer-affected regions based on medical images for cancer diagnosis, additional medical information of the patient that could be used for diagnosis should not be directly used for instance prediction according to Theorem 9-11 and Condition 9, but should be used as weights according to Theorem 9.
>
> Your comments significantly contribute to enhancing the utility of the paper by considering the future development and practicality of the proposed framework. Thank you once again for your valuable feedback. If you have any further questions or concerns, please let me know so that I can further address them.
>
> **Reference:**
>
> [1] Ilse et al., (2018, July). Attention-based deep multiple instance learning. In *International conference on machine learning* (pp. 2127-2136). PMLR.
>
> [2] Xu et al., (2023). Classification of colorectal cancer consensus molecular subtypes using attention-based multi-instance learning network on whole-slide images. *Acta Histochemica*, *125*(6), 152057.
>
> [3] Han et al., (2020). Accurate screening of COVID-19 using attention-based deep 3D multiple instance learning. *IEEE transactions on medical imaging*, *39*(8), 2584-2594.
>
> [4] Li et al., (2020, December). Deep multi-instance learning with induced self-attention for medical image classification. In *2020 IEEE International Conference on Bioinformatics and Biomedicine (BIBM)* (pp. 446-450). IEEE.

---

> > ### Comment · Reviewer_dSpL · 2024-08-13
> >
> > I thank the authors for their response, it has made a good case about how the paper fits into the broader field - this analysis would be a good addition to the paper for a camera ready version should the paper be accepted. My concerns have been addressed. I am bumping my score by one more point.

---

> > > ### Author Response · Authors · 2024-08-13
> > >
> > > We deeply appreciate your positive evaluation of our paper and your kind acknowledgment of our responses. We are pleased to have addressed all your concerns. We will incorporate your suggestions into the final camera-ready version, and we believe that this will enhance the contribution and quality of the paper. Thank you once again for your valuable feedback, and please do not hesitate to reach out if you have any additional questions or concerns.

---

### Official Review · Reviewer_2msW · 2024-07-13

**Soundness:** 3
**Presentation:** 3
**Contribution:** 3
**Rating:** 6
**Confidence:** 2

**Summary:**

The paper addresses a critical gap in Multiple Instance Learning research by proposing a theoretical framework to evaluate instance-level learnability of MIL algorithms. Utilizing PAC learning theory, the authors derive conditions under which Deep MIL algorithms can achieve instance-level learnability. The framework is then applied to evaluate the learnability of several existing Deep MIL algorithms, and the findings are validated through empirical studies.

**Strengths:**

1. The paper introduces a novel theoretical framework to assess instance-level learnability in MIL, filling a significant gap in current research.
2. The authors provide clear and precise definitions of MIL domain spaces, hypothesis spaces, and risk, which aids in understanding the theoretical constructs.
3. The theoretical conditions and their proofs are well-articulated and grounded in PAC learning theory, providing a strong foundation for the proposed framework.
4. The paper evaluates a range of Deep MIL algorithms against the proposed theoretical conditions, offering a broad validation of the framework.
5. The study addresses practical concerns in domains requiring expert labeling, such as pathology, highlighting the potential impact of instance-level learnable MIL algorithms.

**Weaknesses:**

1. The paper’s theoretical sections are dense and may be challenging for readers not well-versed in PAC learning theory.
2. The framework relies on several assumptions, such as the independence of instances in certain conditions, which may not always hold in real-world applications.
3. Details on experimental setups and reproducibility of empirical studies are sparse, which could hinder replication efforts.

**Questions:**

1. How do the proposed theoretical conditions translate to improvements in practical MIL applications? Can you provide more detailed real-world examples?
2. How robust are the theoretical conditions under different assumptions, such as dependent instances within bags?
3. Can the framework be extended to other types of machine learning algorithms beyond Deep MIL?

---

> ### Author Rebuttal · Authors · 2024-08-07
>
> We appreciate your recognition and feedback on our work. We've integrated your suggestions to enhance our manuscript. For more information, see Global Response. We are addressing your specific comments as follows:
>
> **(W1)** Complexity of the Theoretical Framework
>
> - To facilitate understanding of the proposed theoretical framework, we will add a summary figure and explanation of the relationships between the theorems to the paper. This figure, shown as Figure A in the Global Rebuttal PDF, will illustrate how the theorems are derived and their implications.
> - Figure A explains which theorems are derived from others and their outcomes, particularly whether certain pooling methods are learnable at the instance level. This should help readers unfamiliar with PAC learning theory to understand the goals and theoretical flow of our paper.
>
> **(W2)** Applicability of the Theoretical Framework to Real-World Applications
>
> - We addressed the learnability of MIL algorithms in independent scenarios because many studies assume independence when implementing MIL algorithms. Most studies using instance-pooling require instances to be independent.
> - To assess the learnability of algorithms in broader scenarios, we defined the General Bag Domain Space, which includes both independent and dependent bag domain spaces. This ensures that our framework is valid under Assumption 1, even when relationships between instances exist.
>
> **(W3)** Limited Empirical Validation
>
> - The experiments in our paper are reproducible, with code made available and a detailed readme file explaining the experimental setup and parameters. Appendix D provides extensive details on the experimental environment, hyperparameters, and network structures to facilitate replication.
> - Additionally, reflecting reviewers' suggestions for more empirical validation with recent Deep MIL algorithms, we conducted further experiments detailed in item 1 of the Global Rebuttal, shown in Tables A and B. These results will be added to the paper, demonstrating that our theoretical framework applies to all Deep MIL algorithms, not just representative pooling methods, and confirming its empirical validity.
>
> **(Q1)** Practical Applications of the Framework
>
> - Our proposed framework offers the advantage of theoretically evaluating whether MIL algorithms are learnable at the instance level. The Global Rebuttal PDF's Figure A clearly demonstrates the theoretical learnability of individual pooling methods, and we have empirically validated these theories through experiments on specific pooling algorithms, as shown in Global Rebuttal item 1.
>     - For example, in the medical domain, MIL algorithms have been adopted to address labeling issues. These algorithms often use attention pooling to reflect current trends. However, our study shows that attention pooling is not learnable at the instance level, indicating it is unsuitable for medical applications where accurate instance-level predictions are crucial for patient diagnosis.
>     - Our study identifies two pooling methods that are learnable at the instance level for various MIL domain spaces. This can help select appropriate MIL algorithms for specific environments.
> - Additionally, our framework can be applied to individual techniques aimed at improving MIL performance.
>     - In Section 4.4.3, we applied our framework to positional encoding, commonly used in MIL research, showing in Table 4 that positional information should be used only in attention computation to improve instance-level prediction performance.
>     - As shown in Table C of the Global Rebuttal, our framework also theoretically guarantees the best performance for models that use relationships between instances from different bags, such as in MD-MIL scenarios.
>
> **(Q2)** Theoretical Robustness of the Framework in the General Bag Domain Space
>
> - We proposed a framework under Assumption 1, which assumes MIL algorithms are learnable at the bag level, to facilitate the theoretical evaluation of instance-level learnability. Under this assumption, our framework includes both the Independent bag domain space ($D_{XY}^{Ind}$) and Dependent bag domain space ($D_{XY}^{Dep}$) in the General Bag Domain Space ($D_{XY}^{Gen} = D_{XY}^{Ind} \cup D_{XY}^{Dep}$) and proposes Theorem 3 to determine instance-level learnability. Based on Theorem 3, we derived Theorems 5, 8, 9, 10, and 11.
> - Specifically, Theorem 3 is proven using the Azuma-Hoeffding inequality [1], which applies to dependent random variables, demonstrating the robustness of our theoretical conditions. Empirical validation of our theorems confirms the robustness of the framework in $D_{XY}^{Gen}$.
>
> **(Q3)** Extending the Framework to Other Types of Machine Learning Algorithms
>
> - Our framework can be applied to machine learning-based MIL algorithms, provided they satisfy Assumption 1.
>     - For example, the well-known machine learning-based MIL algorithm MI-SVM [2] performs individual instance predictions and then predicts the bag based on these results, satisfying Condition 4 and being learnable at the instance level.
> - Recent MIL research focuses more on Deep MIL than on machine learning-based methods because deep networks can better extract important features from instances, achieving higher performance [2]. Thus, we refer to our framework as a framework for Deep MIL.
>
> **Reference:**
>
> [1] Pelekis et al., Hoeffding’s inequality for sums of dependent random variables. *Mediterranean Journal of Mathematics*, 2017
>
> [2] Andrews et al., Support vector machines for multiple-instance learning. Neurips, 2002

---

> > ### Comment · Reviewer_2msW · 2024-08-14
> >
> > Thanks for the response. The reply is appreciated. I hence raised my score to 6.

---

> > > ### Author Response · Authors · 2024-08-14
> > >
> > > Thank you for your positive evaluation of our paper and for raising your score. Your feedback has been invaluable in enhancing our research. If you have any further concerns or questions, please do not hesitate to let us know. We are more than happy to address them.

---

### Author Rebuttal · Authors · 2024-08-07

Thank you for your valuable feedback and insightful comments. We appreciate the time and effort you have invested in reviewing our work. K1fQ and W4rs recognized our focus on the theoretical foundations of our study. dSpL and 2msW acknowledged the clarity and rigor of our theoretical framework and arguments. Finally, bVsY appreciated the clarity and potential impact of our work within the MIL community.

We did our best to address all of your comments and will make the necessary revisions to improve the clarity, rigor, and validity of our manuscript. We kindly ask for your consideration in adjusting the review score if you find that our response has effectively addressed your concerns. In the revised manuscript, we will make the following major modifications.

1. **Extensive Expansion of Experimental Validation**
    - We have expaned experimental validation to a wider range of Deep MIL algorithms to demonstrate the practical applicability of our theoretical framework.
    - Tables A and B of the attached PDF show experimental validation results for Deep MILs, which confirm the same conclusions as the theoretical framework of this paper. We will include detailed analysis and experimental results for these conclusions in the revised manuscript.
    - Based on experimental validation, it was confirmed that MIL algorithms using the same pooling method exhibit a similar overall trend to the theoretical analysis of the framework, despite slight performance differences.
2. **Practical usage of Multi-Dimensional MIL (MD-MIL)**
    - MD-MIL involves learning from bag-of-bags labels, predicting labels for the bags that comprise these bag-of-bags, and for the instances within each bag.
    - According to Theorems 9, 10, and 11, using information from instances outside the target instance for prediction should be limited to the attention mechanism. Thus, in predicting instances within a specific bag, the relationships with instances from other bags should be utilized only through attention operations.
    - As the research on MD-MIL is still in its early stages, this study does not consider all possible pooling combinations for MD-MIL. Instead, it aims to demonstrate the practical effectiveness of MD-MIL by investigating whether a model that theoretically considers the relationships between bags can directly enhance performance. To this end, experiments on MD-MIL were conducted using the ShanghaiTech Video Anomaly Detection (VAD) dataset.
    - We evaluated the prediction performance of frames (i.e., bags) and image patches (i.e., instances) in videos, comparing three methods according to the instances used from other bags for instance prediction: 1) **None-Attention,** which does not utilize attention mechanisms, 2) **Attention**, which uses attention mechanisms within the same bag, and 3) **Cross-Attention**, which uses attention mechanisms that leverage features from instances in other bags.
        - These three attention mechanisms are theoretically used for **Conjunctive Pooling** in the **General Domain Space**, at each dimension (instance and bag).
    - The experimental results are in Table C of the attached PDF. The Cross-Attention method showed the highest performance by effectively utilizing relationships between instances across different bags, confirming the practical usage of MD-MIL.
3. **Overall presentation enhancement**
    - We will add the figures in the attached PDF to the paper to help readers understand.
        - **Figure A**: A summary diagram showing the relationships between theorems. This will help readers understand the flow and connections between the theorems and the corresponding pooling methods.
    - We believe these revisions address the reviewers' concerns and enhance the overall quality and clarity of the manuscript. Thank you for considering our response.

**References:**

[1] Shi et al., Loss-based attention for deep multiple instance learning. AAAI, 2020.

[2] Liu et al., Weakly-Supervised Residual Evidential Learning for Multi-Instance Uncertainty Estimation. ICML, 2024.

[3] Wang et al., Rethinking Multiple Instance Learning for Whole Slide Image Classification: A Bag-Level Classifier is a Good Instance-Level Teacher. IEEE Transaction on Medical Imaging, 2024.

[4] Shao et al., Transmil: Transformer based correlated multiple instance learning for whole slide image classification. Neurips, 2021.

[5] Zhang et al., Multi-instance causal representation learning for instance label prediction and out-of-distribution generalization. Neurips, 2022.

[6] Rymarczyk et al., Kernel self-attention in deep multiple instance learning. arXiv, 2020.

[7] Wang et al., Revisiting multiple instance neural networks. Pattern recognition, 2018.

[8] Ilse et al., Attention-based deep multiple instance learning. PMLR., 2018.

[9] Javed et al., Additive mil: Intrinsically interpretable multiple instance learning for pathology. Neurips, 2022.

[10] Early et al., Inherently interpretable time series classification via multiple instance learning. ICLR, 2024.

---

### Decision · Program_Chairs · 2024-09-25

**Decision:**

Accept (poster)

**Comment:**

The paper develops a novel theoretical framework to investigate the instance-level learnability of deep multiple instance learning based on the probably approximately correct (PAC) learning theory. The proposed framework fills out a critical gap to understand existing MIL algorithms through the lens of PAC learning theory. The theories along with their conditions can be leverages to choose the most appropriate MIL algorithm based on the specific settings of the problem. They also provide useful insights that could help to shape the future MIL research.

All the reviewers participated in the discussions with the authors. The authors did a decent job in their rebuttal and during their follow-up discussions with the reviewers. In the end, all reviewers assigned a positive rating in support of the acceptance of the paper. The authors are encouraged to include the important comments/suggestions from the reviewers into the final version of the paper.